# LaViDa-R1: Advancing Reasoning for Unified Multimodal Diffusion Language Models

Shufan Li[1,2,*,†], Yuchen Zhu[1,3,*,†], Kangning Liu[1], Zhe Lin[1]
Yongxin Chen[3], Molei Tao[3], Aditya Grover[2], Jiuxiang Gu[1], Jason Kuen[1]
[1]Adobe  [2]UCLA  [3]Georgia Tech
* Equal Contribution   † Work done primarily during internship at Adobe Research

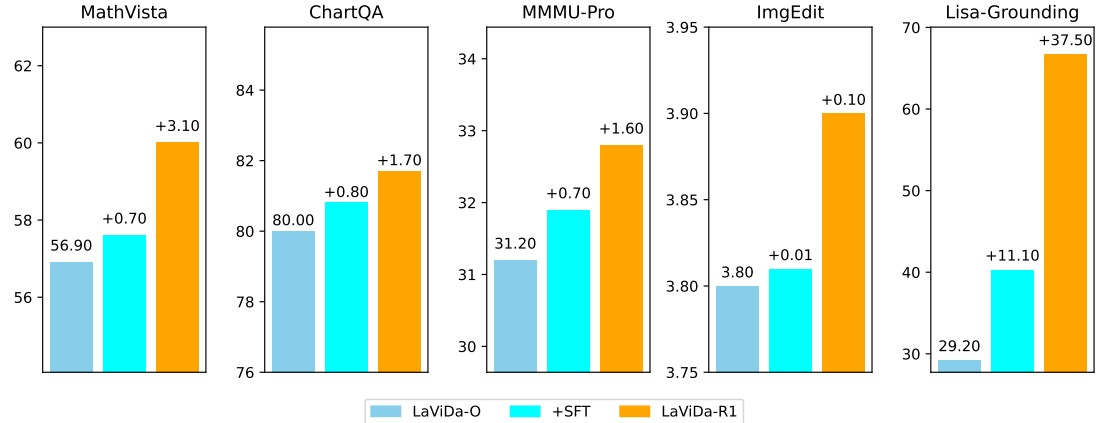

*Figure 1.* **We introduce LaViDa-R1**, a multimodal diffusion language model with strong reasoning capabilities across diverse tasks. LaViDa-R1 incorporates a novel unified post-training that significantly improves upon the base model LaViDa-O (Li et al., 2025b) and SFT baseline on visual math reasoning, visual question answering, image editing, and object grounding tasks.

## Abstract

Diffusion language models (dLLMs) recently emerged as a promising alternative to auto-regressive LLMs. The latest works further extended it to multimodal understanding and generation tasks. In this work, we propose LaViDa-R1, a multimodal, general-purpose reasoning dLLM. Unlike existing works that build reasoning dLLMs through task-specific reinforcement learning, LaViDa-R1 incorporates diverse multimodal understanding and generation tasks in a unified manner. In particular, LaViDa-R1 is built with a novel unified post-training framework that seamlessly integrates supervised finetuning (SFT) and multi-task reinforcement learning (RL). It employs several novel training techniques, including answer-forcing, tree search, and complementary likelihood estimation, to enhance effectiveness and scalability. Extensive experiments demonstrate LaViDa-R1's strong performance on a wide range of multimodal tasks, including visual math reasoning, reason-intensive grounding, and image editing.

## 1. Introduction

Unified Multimodal Large Language Models (MLLMs) such as GPT-4o (OpenAI, 2024) have demonstrated strong utility on diverse scenarios. Traditionally, these models are built as auto-regressive (AR) models that generate tokens sequentially. Recently, dLLMs have emerged as a promising alternative to auto-regressive models in many language (Nie et al., 2025; Ye et al., 2025a) and multimodal tasks (Yang et al., 2025b; Li et al., 2025b;c; Swerdlow et al., 2025). Instead of generating tokens in a left-to-right order, dLLMs start with a fully masked sequence and gradually unmask it through multiple diffusion steps, decoding multiple tokens in parallel. Compared with AR models, dLLMs offer many

. Correspondence to: Shufan Li <jacklishufan@cs.ucla.edu>, Yuchen Zhu <yzhu738@gatech.edu>, Jiuxiang Gu <jigu@adobe.com>, Jason Kuen <kuen@adobe.com>.

*Proceedings of the 43ʳᵈ International Conference on Machine Learning*, Seoul, South Korea. PMLR 306, 2026. Copyright 2026 by the author(s).

attractive properties such as faster inference speed (Wu et al., 2025b), bi-directional context (Li et al., 2025c; Nie et al., 2025), and a unified generation paradigm for visual and text tokens (Li et al., 2025b; Yang et al., 2025b).

To improve the performance of AR models, a common technique is to incorporate a reasoning process (Guo et al., 2025), in which the model generates text-reasoning traces before producing the final output. This approach has shown to be highly effective on complex tasks such as math reasoning (Shao et al., 2024) and coding (Li et al., 2025a). Recent work (Shen et al., 2025; Deng et al., 2025a) further extends the reasoning process to support multimodal understanding and generation tasks.

The reasoning capability of a model is typically acquired via post-supervised finetuning on Chain-of-Thought (CoT) data (Wei et al., 2022) and reinforcement learning (RL) (Guo et al., 2025; Yang et al., 2025a). While these approaches were first developed for AR models, recent work has explored applying them to build reasoning dLLMs and multimodal dLLMs (Yang et al., 2025b; Zhao et al., 2025a; Zhu et al., 2025). While AR models suffer from linear error accumulation due to causal masking, reasoning dLLMs leverage global visibility. By jointly modeling reasoning and results, they enable holistic refinement, allowing the emerging answer to adjust the reasoning trace and providing a unified framework effective for both spatial visual modeling and complex logical reasoning.

While the reasoning dLLM literature has seen some progress, several key challenges remain unaddressed. First, existing work focuses on a limited set of tasks, such as mathematical reasoning, and often requires dataset-specific fine-tuning. Extending RL to build a general-purpose reasoning model that supports diverse, multimodal tasks such as image editing and reason-intensive object grounding remains largely unexplored. Second, training dLLM with reinforcement learning is prone to collapse even in the presence of KL divergence regularizer. Furthermore, incorporating the KL-regularizer hinders models' exploration during training, thereby degrading task performance. Third, for complex or difficult tasks, the model may fail to generate high-quality samples during training, leading to a low-quality training signal or, in the worst case, a zero training signal due to diminishing returns. Finally, unlike AR models, which can evaluate sequence likelihood exactly and efficiently, computing sequence likelihood for dLLMs is intractable and is typically estimated via the Monte Carlo (MC) method. This approach poses unique challenges for training stability as it produces missing and imbalanced token gradients.

To address these gaps, we propose LaViDa-R1, a recipe for building strong-performing multimodal dLLMs. Compared with existing methods, LaViDa-R1 introduces several key innovations. First, it introduces a unified framework that encompasses a diverse range of visual and language tasks, including mathematical reasoning, visual question answering, reason-intensive grounding, and image editing. Second, it introduces a novel post-training objective that seamlessly integrates SFT and the RL paradigm. By replacing the KL divergence term with SFT regularization, LaViDa-R1 allows the model to sufficiently explore beyond the distribution of a pretrained base model while also preventing collapse. Third, to address the lack of a training signal when no high-quality samples are generated for difficult prompts, we incorporate two guided rollout generation mechanisms to construct high-quality samples. When ground-truth answers are available, we employ an answer-forcing mechanism that leverages dLLMs' inpainting capabilities to artificially construct high-quality reasoning traces on the fly. When the ground-truth answer is unavailable, we employ a tree-search algorithm that tailors the generated distribution towards higher-quality outputs. Finally, we propose a complementary likelihood estimator that improves upon existing MC methods by addressing the missing-signal and imbalanced-gradient problems discussed above.

To validate the effectiveness of LaViDa-R1, we conducted extensive experiments covering a wide range of tasks. Results show that LaViDa-R1 achieves strong reasoning performances on multiple benchmarks such as MathVerse, ChartQA, Lisa-Grounding, and ImgEdit.

## 2. Background and Related Works

### 2.1. Discrete diffusion models

Early works on discrete diffusion models (Austin et al., 2021; Sahoo et al., 2024; Lou et al., 2023; Ou et al., 2024; Shi et al., 2024) first developed principled frameworks for training and sampling from masked generative models (MGMs) by formalizing the unmasking process of MGMs as a discrete diffusion process. Later works, such as Mercury and LLaDA (Khanna et al., 2025; Nie et al., 2025; Ye et al., 2025a), scaled discrete diffusion models to large-scale language modeling, achieving performance comparable to autoregressive LLMs while offering benefits such as bidirectional context and faster inference. Recent works such as LaViDa-O and MMaDa (Li et al., 2025c;b; Yu et al., 2025b; Shi et al., 2025; Yang et al., 2025b) further expanded dLLMs to multimodal understanding and generation tasks.

Formally, given a sequence $\boldsymbol{y}_0$ of length $L$, and a conditional prompt $\boldsymbol{x}$, the forward masked diffusion process $q(\boldsymbol{y}_t|\boldsymbol{y}_s)$ progressively mask tokens over the time interval $[0, 1]$, with $1 \geq t \geq s \geq 0$. At $t = 0$, no tokens are masked. At $t = 1$, the sequence $\boldsymbol{y}_1 = [\mathrm{M}, \mathrm{M}, \cdots, \mathrm{M}]$ consists entirely of a special mask token M. When $0 < t < 1$, $\boldsymbol{y}_t$ contains a mixture of clean and masked tokens. A dLLM policy model $\pi_\theta$ is trained to model the reverse process $p(\boldsymbol{y}_s|\boldsymbol{y}_t, \boldsymbol{x})$. The

masked diffusion objective is defined as:

$$\mathcal{L}_{\text{dLLM}}(\theta) = -\mathbb{E}_{t,\boldsymbol{y}_0,\boldsymbol{y}_t,\boldsymbol{x}} \left[ \frac{1}{t} \log \pi_\theta(\boldsymbol{y}_0|\boldsymbol{y}_t,\boldsymbol{x}) \right] \quad (1)$$

where $\pi_\theta(\boldsymbol{y_0}|\boldsymbol{y_t},\boldsymbol{x})$ is factorized to the product of per-token distribution $\prod_{i=1}^{L} \pi_\theta(\boldsymbol{y_0}[i]|\boldsymbol{y_t},\boldsymbol{x})$ (Sahoo et al., 2024). At inference, given a prompt $\boldsymbol{x}$, we initialize with a fully masked sequence $\boldsymbol{y}_1$ and iteratively apply the learned reverse process $\pi_\theta$ to progressively unmask tokens until a clean sequence $\boldsymbol{y}_0$ is obtained.

## 2.2. Reinforcement Learning

Reinforcement learning (Schulman et al., 2017) can effectively improve the reasoning capability of LLMs. GRPO (Shao et al., 2024) is one of the best-performing RL methods, whose objective has the following form

$$J_{\text{grpo}}(\theta) = \mathbb{E}\Big[ \frac{1}{N} \sum_{i=1}^{N} \min\Big( \frac{\pi_\theta(\boldsymbol{y}^i|\boldsymbol{x})}{\pi_{\text{old}}(\boldsymbol{y}^i|\boldsymbol{x})} A_i^{\text{GRPO}},$$

$$\text{clip}\Big( \frac{\pi_\theta(\boldsymbol{y}^i|\boldsymbol{x})}{\pi_{\text{old}}(\boldsymbol{y}^i|\boldsymbol{x})}, 1-\varepsilon, 1+\varepsilon \Big) A_i^{\text{GRPO}} \Big) - \beta \, \text{kl}(\boldsymbol{y}^i) \Big],$$

$$A_i^{\text{GRPO}} = \frac{r_i - \text{mean}(r_1,\dots,r_N)}{\text{std}(r_1,\dots,r_N)} \quad (2)$$

where $\boldsymbol{y}^1,\dots,\boldsymbol{y}^N$ is a group of $N$ responses sampled with prompt $\boldsymbol{x}$, $\text{kl}(\boldsymbol{y}^i)$ is a per-sample reverse KL estimator, $r_i$ is the per-sample reward and $A_i^{\text{GRPO}}$ is the per-sample advantage, which is the normalized reward.

It has been shown that in a pure on-policy setting, GRPO advantages can be simplified as

$$J(\theta) = \mathbb{E}\Big[ \frac{1}{N} \sum_{i=1}^{N} A_i^{\text{GRPO}} \log \pi_\theta(\boldsymbol{y}^i|\boldsymbol{x}) - \beta \, \text{kl}(\boldsymbol{y}^i) \Big] \quad (3)$$

Interestingly, it is shown that by changing $\beta$, and the definition of $A_i$ (Shao et al., 2024), we can use the same form to represent many other objectives such as SFT and DPO (Rafailov et al., 2023). Inspired by this view, our work proposes a practical unified post-training method that combines SFT, RL, and self-distillation loss to improve reasoning capabilities.

**RL for dLLMs.** Multiple works have also explored applying GRPO-style RL to dLLMs (Zhao et al., 2025a; Gong et al., 2025; Wang et al., 2025a; Tang et al., 2025; Zhu et al., 2025; Wang et al., 2025e; Ou et al., 2025; Zheng et al., 2025a), mostly focused on language-only tasks with task-specific training. Few works have explored applying RL to multimodal tasks. Uni-GRPO (Yang et al., 2025b) first explored extending RL to improve math reasoning, image captioning and text-to-image generation simultaneously.

Our work extends RL to a broader set of tasks such as reason-intensive object grounding and image editing. We provide a more thorough review to these literature in Appendix D.

**RL for Multimodal tasks.** For AR VLMs and unified MLLMs, many works explored enhancing reasoning with RL (Wang et al., 2025b; Shen et al., 2025; Meng et al., 2025; Zhou et al., 2025; Yang et al., 2025c; Deng et al., 2025b; Huang et al., 2025; Wang et al., 2025c; Yuan et al., 2025), achieving successes on a wide range of visual understanding and generation tasks. We also note that while RL is commonly associated with reasoning, it can also be applied to improve visual generation tasks without reasoning elements. Multiple works explored applying RL in image-output-only setup (e.g. Stable Diffusion) to improve text-to-image generation and image-editing quality (Li et al., 2025d; Geng et al., 2025; Wei et al., 2025; Liu et al., 2025a; Zheng et al., 2025b; Wu et al., 2025a; Luo et al., 2025b). Most existing work focuses on applying RL techniques in a task-specific manner, with a few exceptions exploring unified reasoning for unified multimodal models (Tian et al., 2025; Xin et al., 2025; Yang et al., 2025b; Cui et al., 2025). Our work focuses specifically on improving multimodal task performance by enhancing *reasoning* capabilities using RL. It is more closely aligned with the VLM and MLLM reasoning literature than with the general RL literature on visual generation.

## 2.3. Improve reasoning with non-online-RL methods

Other lines of work address the challenge of LLM alignment beyond GRPO-style policy-gradient methods. Direct Preference Optimization (DPO) (Rafailov et al., 2023; Azar et al., 2024; Zhao et al., 2023) aligns LLMs using off-line paired preference data, and Online-DPO (Guo et al., 2024b) optimizes models based on preference pairs obtained from model-generated responses and an external reward model as a judge. Self-play (Chen et al., 2024; Wu et al., 2024; Rosset et al., 2024; Swamy et al., 2024) formulate the task of LLM alignment as a two-player game. Self-distillation-type methods (Yang et al., 2024; Amini et al., 2024) progressively improve the model by distilling optimized self-generated rollouts back into the model.

Specifically, BOND (Sessa et al., 2024) iteratively distilled the Best-of-N model output distribution into the model policy by minimizing the KL divergence between them, which is equivalent to performing SFT on the best sequence $\boldsymbol{y}^j$ among a group of responses $\boldsymbol{y}^1,\dots,\boldsymbol{y}^N$ from the same prompt $\boldsymbol{x}$, with the following objective

$$J_{\text{distill}}(\theta) = \log \pi_\theta(\boldsymbol{y}^j|\boldsymbol{x}), \; j = \text{argmax}_i \; r_i \quad (4)$$

Our work creatively combines the best-of-N distillation objective with the standard RL objective. We also explored alternatives like Online-DPO. Further discussions can be

found in Appendix A.2.

# 3. Method

In this section, we introduce the training framework for LaViDa-R1, which comprises three main components: a unified post-training policy-gradient objective, guided rollout-generation algorithms that efficiently sample high-quality outputs during training, and a novel, stable, complementary-masking-based likelihood estimator.

## 3.1. Unified Post-training

As is first noted in (Shao et al., 2024), many post-training objectives, including online GRPO (Shao et al., 2024) (without KL regularization), Online DPO (Guo et al., 2024b), SFT, and self-distillation (Sessa et al., 2024), can be written in the same form of policy gradient objectives:

$$J_{\text{Unified}}(\theta) = \frac{1}{N} \sum_{i=1}^{N} A_i \log \pi_\theta(\boldsymbol{y}^i | \boldsymbol{x}^i) \quad (5)$$

The design choices that differentiate these objectives are the sources of $(\boldsymbol{y^i}, \boldsymbol{x^i})$ pairs and the per-sample weights $A_i$. For example,

- When $\boldsymbol{y}^i \sim \pi_\theta(\cdot|\boldsymbol{x}^i)$ is sampled form the policy model during training, $A_i = A_i^{\text{GRPO}}$, $J(\theta)$ is the online GRPO objective with zero KL regularization from Eq. 3.

- When $\boldsymbol{y}^i \sim \pi_\theta(\cdot|\boldsymbol{x}^i)$, $A_i = A_i^{\text{distill}}$, with $A_i^{\text{distill}} = 1$ if $i = \arg\max(r_1, \ldots, r_N)$ and 0 otherwise, $J(\theta)$ is equivalent to the best-of-N distillation (Eq. 4).

- When $(\boldsymbol{y}^i, \boldsymbol{x}^i) \sim \mathcal{D}$ comes from a offline training dataset, $A_i = A_i^{\text{sft}} = 1$, $J(\theta)$ is the SFT loss.

This observation has several important implications. First, we can easily combine training batches of these objectives by concatenating lists of $(\boldsymbol{y}^i, \boldsymbol{x}^i)$ and corresponding $A_i$. This pipeline is concretely shown in Figure 2. At each training step, a generic data engine provides pairs of prompts and responses $(\boldsymbol{y}^i, \boldsymbol{x}^i)$, as well as corresponding advantage values $A_i$. These can be obtained either by loading from an offline training dataset or by online generation, followed by reward and advantage calculation such as $A_i^{\text{distill}}$ or $A_i^{\text{GRPO}}$. The policy model is then used to compute the log-likelihood of each sequence $\log \pi_\theta(\boldsymbol{y}^i | \boldsymbol{x}_i)$. Finally, we optimize the unified objective in Eq. 5. This design is illustrated in Fig 2.

Second, for on-policy objectives where $\boldsymbol{y}^i \sim \pi_\theta(\cdot|\boldsymbol{x}^i)$ are sampled from the policy model, we can efficiently combine different objectives by simply aggregating the advantage values from each method using a weighted average, without the need to resample rollouts across different losses.

For example, we can simultaneously perform online GRPO and best-of-N self-distillation by adopting a new advantage $A_i^{\text{aggr}} = \gamma A_i^{\text{distill}} + (1-\gamma) A_i^{\text{GRPO}}$ for each sample with barely any additional computational overhead.

In our final design, we combined SFT, online GRPO and online self-distillation objectives, with $\gamma = 0.5$. We also explored other objectives that can be written in this form such as online DPO and SLiC (Rafailov et al., 2023; Zhao et al., 2023). Further details are provided in Appendix A.2.

Intuitively, adding the SFT objective can serve as a substitute for KL regularization. It allows the model to sufficiently explore the action space without being constrained by a suboptimal reference model, while preventing collapse. Furthermore, from a computational-efficiency perspective, removing the need for a reference model significantly reduces the cost of RL training, since we no longer need to load it into GPU memory or host it on a separate server. On the other hand, incorporating a self-distillation objective amplifies the training signal from the best sample in the group, leading to stronger training signals.

## 3.2. Guided Rollout Generation

Online RL with group-based advantage computation is known to suffer from a vanishing training signal when all generated responses receive low rewards, resulting in zero advantage for all responses and rendering the RL process ineffective. To effectively address this notorious issue, we propose using guided generation to create high-quality rollout samples during training. We consider two types of guided generation algorithms, each with different operating scenarios. Answer-forcing is applied when we have access to the ground-truth answers to the training questions (e.g. math reasoning). When answers are unavailable, we resort to tree search, which is applicable when a real-valued reward function is available.

### 3.2.1. ANSWER FORCING

We leverage dLLMs' bidirectional generation capabilities to construct high-quality reasoning traces when ground truth answers are available. When the policy model fails to generate high-quality outputs in a group (i.e., no correct math solution or no high-IoU bounding boxes), we manually insert the ground truth answer token to the end of a fully masked sequence and leverage dLLM's text-infilling capabilities to inpaint intermediate reasoning traces that lead towards the final answer. We name this guided generation approach **Answer Forcing**. An example is shown in Fig. 3.

Formally, given a prompt $\boldsymbol{x}$, we first sample group responses $\boldsymbol{y}^i, \cdots, \boldsymbol{y}^N \sim \pi_\theta(\boldsymbol{y}|\boldsymbol{x})$. Each $\boldsymbol{y}^i$ typically starts with a text reasoning trace enclosed by "$<$think$>$...$</$think$>$" tags followed by the final answer enclosed in "$<$answer

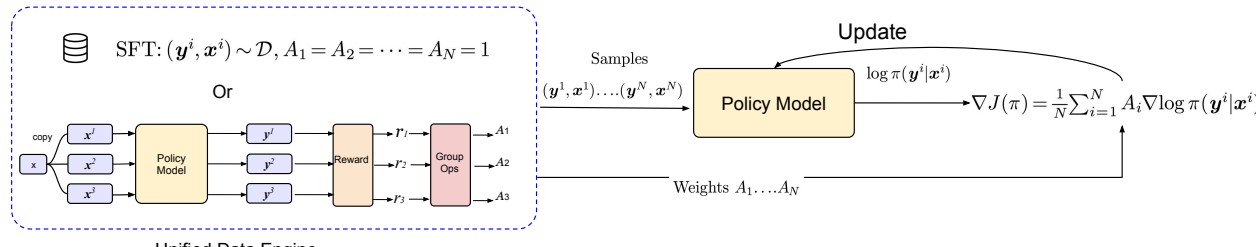

Figure 2. **Unified Post Training Framework of LaViDa-R1.** At each training step, a generic data engine provides prompts-response pairs of $(\boldsymbol{y}^i, \boldsymbol{x}^i)$, and sample weights $A_i$, either by loading from a dataset or by online generation. The policy model is then used to compute the log-likelihood of each sequence $\log \pi_\theta(\boldsymbol{y}^i | \boldsymbol{x}_i)$. Finally, we optimize the proposed unified policy gradient objective.

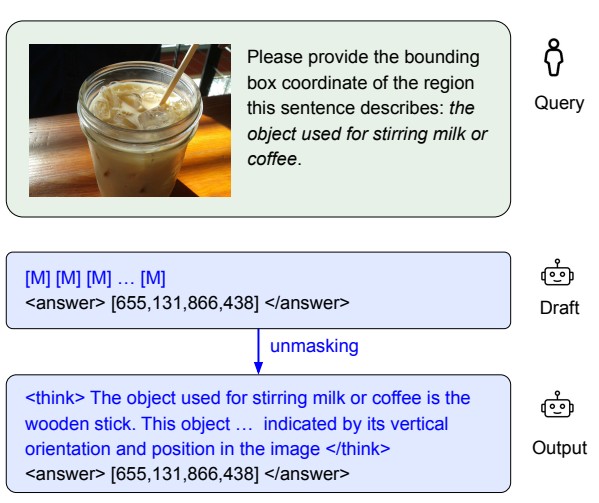

Figure 3. **Answer-Forcing.** We initialize a partially masked sequence with ground truth answer injected at the end, and use the diffusion unmasking process to obtain the reasoning trace.

$> ... </$ answer $>$" tags. The final answer can be either text or image tokens, depending on the tasks.

If all of the extracted answers have low rewards (e.g., incorrect for math reasoning tasks), and a ground truth answer $z^*$ is accessible, we can initialize a new sequence $\boldsymbol{y}^{N+1} =$ "M . . . M $<$ answer $> z^* </$ answer $>$" where M is the special mask token. We then employ the dLLM $\pi_\theta$ to progressively unmask these tokens and effectively generate a synthetic reasoning trace conditioned on the ground-truth answer. This sample is then added to the group. Additional details of answer-forcing are included in Appendix A.4

### 3.2.2. TREE SEARCH

For tasks that do not provide ground-truth answers (such as image editing), we leverage **Tree Search** to obtain high-reward rollouts. Given a base group size of $N$, we first generate $N$ samples and compute rewards as usual. We then find the samples in each group with the highest rewards and generate $N$ new samples, starting from an early state in

the generated trajectories of those samples rather than from fully noised sequences. This gives $N$ new samples. This process is repeated $k$ times, yielding a final effective group size of $Nk$. This process is illustrated in Figure 4.

Concretely, given a prompt $\boldsymbol{x}$, we generate $N$ sequences $\boldsymbol{y}_0^1, \ldots, \boldsymbol{y}_0^N$ through $T$ diffusion steps. We also keep track of intermediate diffusion states $\boldsymbol{y}_{t_i}^1, \ldots, \boldsymbol{y}_{t_i}^N$ where $1 = t_0 > t_1 > ..t_T = 0$ are discretized diffusion timesteps. Notably, $\boldsymbol{y}_{t_0}^1, \ldots, \boldsymbol{y}_{t_0}^N$ are fully masked sequences and $\boldsymbol{y}_{t_T}^1, \ldots, \boldsymbol{y}_{t_T}^N$ are final generated responses, which may contain both image and text tokens. After obtaining rewards $r_1, \cdots, r_N$ for each response, we find an index $m = \text{argmax}(r_1, ... r_N)$ with the highest rewards and retrieve its early diffusion states $\boldsymbol{y}_{t_s}^m$, which is a partially masked sequence. The selection of the timestep $t_s \in \{t_0, ..., t_T\}$, is controlled by a hyperparameter. We then proceed to generate $N$ new samples $\boldsymbol{y}_0^{N+1} .. y_0^{2N}$ using $\boldsymbol{y}_{t_s}^m$ as the initialization as opposed to a fully masked sequence. To generate these samples, we only need to perform $T - s$ diffusion steps. This process is repeated $k$ times until all $Nk$ samples are obtained.

### 3.3. Complementary-Masking Likelihood Estimator

One essential challenge in applying policy gradient methods to dLLMs is the estimation of the data log probability $\log \pi_\theta(\boldsymbol{y}|\boldsymbol{x})$. Unlike AR models whose likelihood has an exact computable form, dLLMs' likelihoods are estimated via the ELBO surrogate. The ELBO for the sequence log probability is expressed as the following formula,

$$\log \pi_\theta(\boldsymbol{y}|\boldsymbol{x}) = \mathbb{E}_{t,\boldsymbol{y}_t}\Big[w(t) \sum_{k \sim M(y_t)} \log \pi_\theta(\boldsymbol{y}[k]|\boldsymbol{y}_t, \boldsymbol{x})\Big]$$

where $w(t)$ is a weighting function and $M(y) = \{k|\boldsymbol{y}[k] = M\}$ is the set of masked indices. The expectation is typically computed via Monte Carlo (MC) Estimator. Existing works on dLLM RL mostly distinguish themselves from others through the choice of $w(t)$ and how they sample $\boldsymbol{y}_t$ in each MC sample. For example, d1 (Zhao et al., 2025a) samples one MC sample at $t = 1$ (i.e. fully-masked sequence), and

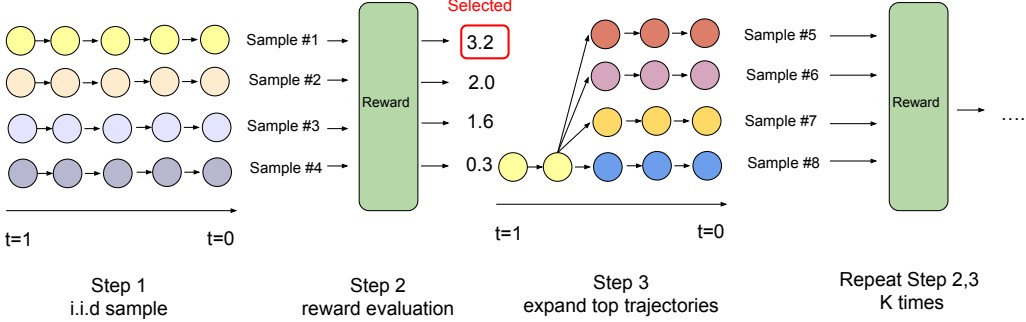

Figure 4. **Tree Search**. Given base group size $N$, we first sample $N$ i.i.d samples and evaluate the rewards. We then select the samples with the highest rewards and generate $N$ new samples from an early diffusion state of the best sample. This process is repeated $K$ times. In this example, $N = 4$.

adopts the weighting $w(t) = 1/t$; UniGRPO (Yang et al., 2025b) samples one MC sample at $t \sim \text{Uniform}([0,1])$ with $w(t) = 1/t$.

In our design, we use two samples with timestep $t_1 \sim \text{Uniform}([0,1])$ and $t_2 = 1-t$. We sample $\boldsymbol{y}_{t_1} \sim q(y_{t_1}|y_0)$ using the discrete forward diffusion process, and set $\boldsymbol{y}_{t_2}[i] = \boldsymbol{y}[i]$ if $\boldsymbol{y}_{t_1}[i] = M$ and $\boldsymbol{y}_{t_2}[i] = M$ if $\boldsymbol{y}_{t_1}[i] \neq M$. This design, known as complementary masking, was first proposed in LaViDa (Li et al., 2025c) for pretraining. For example, if the sequence is $\boldsymbol{y}$ is "there is a dog" and $\boldsymbol{y}_{t_1}$ is "[M] is [M] dog", $\boldsymbol{y}_{t_2}$ will be "there [M] a [M]". We adopt a $w(t) = 1$ instead of $w(t) = \frac{1}{t}$ from LaViDa, giving the following estimator

$$\log \pi_\theta(\boldsymbol{y}|\boldsymbol{x}) = \frac{1}{2} \sum_{j \in \{1,2\}} \sum_{k \sim M(y_{t_j})} \log \pi_\theta(\boldsymbol{y}[k]|\boldsymbol{y}_{t_j}, \boldsymbol{x})$$

Our estimation recipe has several advantages. First, compare with i.i.d MC samples, it masks all tokens once, ensuring the estimate accounts for all tokens in the sequence. This prevents important tokens from being disregarded during training. Second, compare with d1, which can also get estimates for all tokens by always masking every token, it has a smaller training-inference gap. Finally, compared with naively applying complementary masking with $w(t) = \frac{1}{t}$, using $w(t) = 1$ avoids imbalanced token weighting caused by drastically different masking ratios. When $w(t) = \frac{1}{t}$, suppose $t_1 = 0.9$ and $t_2 = 0.1$, we have $\frac{w(t_2)}{w(t_1)} = 9$, indicating that tokens in sample $y_{t_2}$ is $9\times$ more important than those in sample $y_{t_1}$, which is highly unideal since which tokens are masked in which sample is randomly determined.

## 4. Experiments

### 4.1. Setup

We select LaViDa-O as our base model because of its strong multimodal performances and pre-existing reasoning capa-

bilities (Li et al., 2025b). LaViDa-R1 involves two training stages: the first stage is supervised finetuning (SFT) on reasoning data, the second stage is unified post-training on a mix of SFT and RL data using a mix SFT, RL, and self-distillation loss under our unified framework. The RL datasets consist of math reasoning, visual question answering, reason-intensive object grounding and image editing. We use correctness reward for math and QA problems, IoU rewards for object grounding, and the EditScore (Luo et al., 2025b) reward model for image editing. We defer further details on the dataset composition, training schedule, and hyperparameter to Appendix B.

### 4.2. Image Understanding Results

We report results on a wide range of visual understanding tasks and language-only tasks in Table 1. We report results on MathVista (Lu et al., 2023) and MathVerse (Zhang et al., 2024) for visual math reasoning, ChartQA, AI2D and MMMU-Pro (Masry et al., 2022; Kembhavi et al., 2016; Yue et al., 2025) for visual QA and GSM8K and Math500 (Cobbe et al., 2021; Lightman et al., 2023) for language-only tasks. For all datasets, we report the accuracy metric. Notably, we observe that LaViDa-R1 show improvements across all tasks, with the biggest gain coming from the language-only GSM8K and Math500 datasets. We hypothesize that this is because the base model's pretraining dataset is vision-centric, leading to poor language performance and leaving considerable room for improvement.

To further validate the effectiveness of LaViDa-R1, we evaluate on additional multimodal understanding benchmarks that are less reason-intensive such as MMMU (Yue et al., 2024), MMBench (Liu et al., 2024), MME (Fu et al., 2023) and report results in Table 3. Results show that LaViDa-R1 leads to consistent improves. We provide additional qualitative results in Appendix B.5.

*Table 1.* **Performance comparison across visual reasoning, VQA, and language-only benchmarks.** *per-dataset finetuning results. † RL checkpoint for MMada is not open-sourced. Original authors only reported a limited set of results.

| Model | Visual Reasoning | | Visual QA | | | Text Only | |
|---|---|---|---|---|---|---|---|
| | MathVista | MathVerse | ChartQA | AI2D | MMMU-Pro | GSM8K | MATH-500 |
| *Language-Only dLLMs* | | | | | | | |
| LLaDA-8B-Instruct (Nie et al., 2025) | – | – | – | – | – | 78.2 | 36.2 |
| + DiffuGRPO (Zhao et al., 2025a) | – | – | – | – | – | 82.1* | 40.2* |
| Dream-7B(Ye et al., 2025a) | – | – | – | – | - | 77.2 | 39.6 |
| *Visual-Understanding-Only dLLMs* | | | | | | | |
| LaViDa-L (Li et al., 2025c) | 44.8 | 27.2 | 64.6 | 70.0 | 27.1 | – | – |
| Dimple (Yu et al., 2025b) | 42.3 | – | 63.4 | 74.4 | – | – | – |
| *Unified-Understanding-and-Generation dLLMs* | | | | | | | |
| MMaDa-8B-Base (Yang et al., 2025b) | 27.1 | 13.4 | 9.6 | 56.1 | 3.2 | 17.4 | 4.2 |
| +CoT SFT (Yang et al., 2025b) | 33.7 | 13.5 | 9.8 | 66.6 | 8.4 | 65.2 | 26.5 |
| +UniGRPO† (Yang et al., 2025b) | – | – | – | – | – | 73.4 | 36.0 |
| LaViDa-O (Li et al., 2025b) | 56.9 | 36.9 | 80.0 | 76.7 | 31.2 | 47.4 | 23.4 |
| +SFT | 57.6 | 36.6 | 80.8 | 78.6 | 31.9 | 70.6 | 31.0 |
| LaViDa-R1 | **60.0** | **38.7** | **81.7** | **78.9** | **32.8** | **81.5** | **38.6** |

*Table 2.* **Per-Category and overall scores on ImgEdit benchmark.**

| Model | Add | Adjust | Extract | Replace | Remove | Background | Style | Hybrid | Action | Overall |
|---|---|---|---|---|---|---|---|---|---|---|
| GPT-4o (OpenAI, 2024) | 4.61 | 4.33 | 2.90 | 4.35 | 3.66 | 4.57 | 4.93 | 3.96 | 4.89 | 4.20 |
| Qwen2.5VL+Flux (Wang et al., 2025f) | 4.07 | 3.79 | 2.04 | 4.13 | 3.89 | 3.90 | 4.84 | 3.04 | 4.52 | 3.80 |
| FluxKontext dev (Labs et al., 2025) | 3.76 | 3.45 | 2.15 | 3.98 | 2.94 | 3.78 | 4.38 | 2.96 | 4.26 | 3.52 |
| OmniGen2 (Wu et al., 2025c) | 3.57 | 3.06 | 1.77 | 3.74 | 3.20 | 3.57 | 4.81 | 2.52 | 4.68 | 3.44 |
| UniWorld-V1 (Lin et al., 2025) | 3.82 | 3.64 | 2.27 | 3.47 | 3.24 | 2.99 | 4.21 | 2.96 | 2.74 | 3.26 |
| BAGEL (Deng et al., 2025a) | 3.56 | 3.31 | 1.70 | 3.30 | 2.62 | 3.24 | 4.49 | 2.38 | 4.17 | 3.20 |
| Step1X-Edit (Liu et al., 2025c) | 3.88 | 3.14 | 1.76 | 3.40 | 2.41 | 3.16 | 4.63 | 2.64 | 2.52 | 3.06 |
| OmniGen (Xiao et al., 2025) | 3.47 | 3.04 | 1.71 | 2.94 | 2.43 | 3.21 | 4.19 | 2.24 | 3.38 | 2.96 |
| UltraEdit (Zhao et al., 2024) | 3.44 | 2.81 | 2.13 | 2.96 | 1.45 | 2.83 | 3.76 | 1.91 | 2.98 | 2.70 |
| AnyEdit (Yu et al., 2025a) | 3.18 | 2.95 | 1.88 | 2.47 | 2.23 | 2.24 | 2.85 | 1.56 | 2.65 | 2.45 |
| InstructAny2Pix(Li et al., 2023) | 2.55 | 1.83 | 2.10 | 2.54 | 1.17 | 2.01 | 3.51 | 1.42 | 1.98 | 2.12 |
| MagicBrush (Zhang et al., 2023) | 2.84 | 1.58 | 1.51 | 1.97 | 1.58 | 1.75 | 2.38 | 1.62 | 1.22 | 1.90 |
| Instruct-Pix2Pix(Brooks et al., 2023) | 2.45 | 1.83 | 1.44 | 2.01 | 1.50 | 1.44 | 3.55 | 1.20 | 1.46 | 1.88 |
| LaViDa-O (Li et al., 2025b) | 4.04 | 3.62 | 2.01 | 4.39 | 3.98 | **4.06** | 4.82 | 2.94 | 3.54 | 3.71 |
| + Reasoning | 4.11 | 3.67 | 2.04 | 4.40 | 4.05 | 4.00 | 4.75 | 3.10 | 4.04 | 3.80 |
| + SFT | 4.11 | 3.80 | 2.21 | 4.46 | 3.90 | 3.86 | 4.76 | 3.09 | 4.14 | 3.81 |
| LaViDa-R1 | **4.25** | **3.90** | **2.32** | **4.52** | **4.06** | 3.86 | **4.87** | **3.10** | **4.18** | **3.90** |

*Table 3.* **Additional multimodal benchmark results.**

| Model | Capabilities | MMMU | MMB | MME |
|---|---|---|---|---|
| LLaDa-V | Understanding Only | 48.6 | 82.9 | 491 |
| MMaDa | Und & Gen | 30.2 | 68.5 | 242 |
| LaViDa-O | Und & Gen | 45.1 | 76.4 | 488 |
| LaViDa-R1 | Und & Gen | **47.0** | **79.2** | **501** |

*Table 4.* **Performance comparison on Lisa-Grounding Dataset.**

| Model | P@0.5 | mIoU$_{box}$ |
|---|---|---|
| *Specialist Models* | | |
| SegLLM (Wang et al., 2025d) | 61.3 | 55.2 |
| LISA-7B (Lai et al., 2024) | 49.4 | 50.6 |
| *General-purpose VLMs* | | |
| Qwen2.5-VL-7B (Bai et al., 2025b) | 32.0 | 28.7 |
| Qwen3-VL-8B(Bai et al., 2025a) | 62.4 | 56.6 |
| *Reinforcement Learning* | | |
| VLM-R1 (Shen et al., 2025) | 63.1 | – |
| LaViDa-O (Li et al., 2025b) | 29.2 | 26.1 |
| +SFT | 40.3 | 36.9 |
| LaViDa-R1 | **66.7** | **60.0** |

## 4.3. Image Editing Results

We evaluate image editing performance on the ImgEdit benchmark (Ye et al., 2025b), and report the benchmark scores in Table 2. These scores measure both visual quality and prompt compliance via a GPT-4 judge model. We note that the base model LaViDa-O already included some reasoning data in its training pipeline, and has reported image editing performance with reasoning. While SFT leads to a negligible improvement (+0.01) , indicating a performance saturation characteristic of supervised scaling, LaViDa-R1 achieves a significant boost (+0.10). This underscores that our unified RL framework successfully drives exploration beyond the modes learned during supervision. We provide additional qualitative results in Appendix B.5.

## 4.4. Reason-Intensive Grounding

We evaluate reason-intensive grounding on Lisa-Grounding dataset(Lai et al., 2024), and report results in Table 4. We report precision@0.5 (P@0.5) and the mean IoU (mIoU) of bounding boxes. While the base model LaViDa-O exhibits strong grounding performance on simple queries, it performs poorly on Lisa-Grounding which requires com-

Table 5. **Ablation Studies of Answer-Forcing.** *Collapsed

|            | M.Vista | Lisa-Gnd. | Math500 |
|------------|---------|-----------|---------|
| Inject 0%   | 57.8    | 63.1      | 36.2    |
| Inject 10%  | **58.9**| **65.0**  | **38.0**|
| Inject 50%  | 58.0    | 64.2      | 35.4    |
| Inject 100% | 4.1*    | 5.1*      | 4.2*    |

Table 6. **Ablation Studies of Tree search**

| Tree Search Steps | Group Size | ImgEdit |
|-------------------|------------|---------|
| N/A               | 16         | 3.85    |
| N/A               | 32         | 3.84    |
| N/A               | 64         | 3.84    |
| $[0, 8]$          | $16 \times 2$ | **3.90** |
| $[0, 8, 16, 32]$  | $16 \times 4$ | 3.87    |

plex visual reasoning. Compared with this baseline, SFT improves the performance by $+10.8$ mIoU and unified post-training further improves the performance by an additional $+22.1$ mIoU. We provide additional qualitative results in Appendix B.5.

## 5. Ablation Studies

To verify the effectiveness of LaViDa-R1, we conduct additional ablation studies to assess several design choices.

**Answer Forcing.** We investigate the effectiveness of answer forcing and report results in Table 5. We explored answer forcing randomly with probabilities of 0%, 10%, 50%, and 100%. Results demonstrate that 10% forcing has the best overall performance. A high inject ratio leads to collapse because answer-forced samples always receive a high correctness reward, even when their reasoning traces are ill-formed, thereby producing potentially misleading learning signals. This problem is particularly severe when answer forcing always occurs, since it implies that most other samples will likely have a negative advantage due to centering.

**Tree Search.** The hyperparameter that controls the tree search behavior is called restart timestep indices, which is a list of integers specifying the branching steps. For example, given a group size of 16 samples per prompt and a 64-step generation pipeline, a tree search of [0,8] means we first sample 16 outputs independently, each run for 64 steps. We then identify the trajectory corresponding to the sample with the highest reward and branch from its 8th step to generate 16 additional samples. These 16 new samples are initialized from the 8th step of the best-performing sample previously generated, underwriting 56 steps each. Results are shown in Tab. 6. Steps [0,8] is a good choice. Steps [0, 8, 16, 32]

Table 7. **Ablation Studies of Likelihood Estimator**

| #MC | Masking | Lisa-Grounding | ImgEdit |
|-----|---------|----------------|---------|
| 1   | i.i.d   | 61.9           | 3.82    |
| 1   | Full    | 59.2           | 3.77    |
| 2   | i.i.d   | 62.1           | 3.86    |
| 2   | Compl.  | **65.0**       | **3.88**|

yield almost identical performance, because starting from a later diffusion step introduces less uncertainty and does not contribute much.

**Likelihood Estimator.** We investigate the effectiveness of our simple likelihood estimation recipe and report results in Table 7. We explored four approaches. In the first setup (row 1), we randomly mask a subset of tokens and compute the likelihood only over masked positions. This is equivalent to UniGRPO with 1 MC sample. In the second setup (row 2), we mask all tokens and thus compute the likelihood over all tokens, which is equivalent to the d1 setup. In the third setup, 2 i.i.d. MC mask samples were explored. Finally, we report the results of our proposed estimation recipe (row 4). The results show that our method achieves the best performance.

**Self-Distillation Loss.** We experimented with varying $\gamma$, the weight of self-distillation loss described in Section 3.1 and report results in Table 8. The results show that combining two loss functions yields better performance. Intuitively, this loss assigns greater importance to the best-generated samples than standard GRPO. These results highlight the flexibility of the proposed unified paradigm.

Table 8. **Ablation Studies of Self-Distillation Loss**

| $\gamma$        | Effective Objective | ImgEdit |
|-----------------|---------------------|---------|
| $\gamma = 0$    | On-Policy GRPO      | 3.86    |
| $\gamma = 0.5$  | Mixed               | **3.90**|
| $\gamma = 1.0$  | Self-Distillation   | 3.84    |

**Unified Loss.** We finally investigated the effectiveness of the proposed unified loss that combines multiple objectives. We plot the average reward per sample during training. The results are shown in Figure 5. We compare with the standard online GRPO with and without the KL regularizer. Results show that the proposed unified loss with SFT as a regularization term is more stable and yields higher reward. We observe that GRPO diverges even with strong KL regularization $\beta = 0.1$ because the KL estimators only compute divergences on sampled tokens and are not suitable for high-entropy image distributions. Specifically, for most samples, the negative-log-likelihood is above 6 for visual generation and is less than 2 for text generation. The high NLL leads to high variance in the KL term.

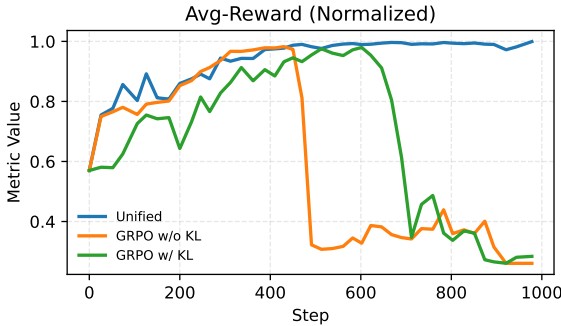

*Figure 5.* **Ablation Studies of Unified Objective.**

## 6. Conclusion

In this paper, we propose LaViDa-R1, a novel training recipe to enhance reasoning ability in unified multimodal dLLMs. LaViDa-R1 introduces a unified post-training paradigm via weighted policy-gradient objectives and a simple yet effective likelihood estimator for stable training. LaViDa-R1 also adopts two new guided rollout generation algorithms to address the key issue of vanishing training signal in online RL. Through multi-task, multi-reward, cross-modality RL, LaViDa-R1 achieves superior performances across a wide range of tasks, including text-only and multimodal reasoning, visual QA, image grounding, and editing.

## Impact Statement

This paper presents work aimed at advancing the field of machine learning. It proposes a unified multimodal model capable of generating text and images, thereby inheriting the full potential of LLMs and image generators. For example, it may be abused to create various harmful and offensive content. We strongly caution the community against such use cases.

## Acknowledgements

AG was supported by NSF CAREER Grant #2341040 and a Schmidt AI 2050 Fellowship. MT is grateful for partial support by NSF Grant DMS-2513699, DOE Grants NA0004261, SC0026274, Richard Duke Fellowship, and Simons Institute for the Theory of Computing at UC Berkeley. YC is grateful for the support from NSF Grant DMS-2450378.

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

# A. Additional Technical Details

## A.1. Formulation of dLLM

In this section, we provide an overview of the standard formulation of dLLMs that are adopted by the literature (Ou et al., 2024; Shi et al., 2024; Sahoo et al., 2024; Lou et al., 2023; You et al., 2025; Li et al., 2025c;b). Notations are adapted from these references to be consistent with the ones used in the main paper to avoid potential confusion.

Given a sequence of discrete tokens $\boldsymbol{y}_0$ whose lengths is $L$, the forward discrete diffusion process $q(\boldsymbol{y}_t|\boldsymbol{y}_s)$ gradually replace the original tokens in $\boldsymbol{y}_0$ to a special mask token $[M]$ over the time interval $[0, 1]$, with $1 \geq t \geq s \geq 0$. At $t = 1$, the sequence $\boldsymbol{y}_1$ is a fully masked sequence. This forward process is formally defined as

$$q(\boldsymbol{y}_t^i|\boldsymbol{y}_s^i) = \begin{cases} \text{Cat}(\boldsymbol{y}_t[i]; \mathbf{M}), & \text{if } \boldsymbol{y}_s[i] = [M] \\ \text{Cat}(\boldsymbol{y}_t[i]; \frac{1-t}{1-s}\mathbf{Y_s}[i] + \frac{t-s}{1-s}\mathbf{M}), & \text{if } \boldsymbol{y}_s[i] \neq [M], \end{cases} \tag{6}$$

where $\text{Cat}(\cdot)$ denotes a discrete categorical distribution, and $\mathbf{M}, \mathbf{Y_s[i]} \in \mathbb{R}^{|V|}$ are probability vectors, and $|V|$ is the vocabulary size. Specifically, $\mathbf{M}$ is a one-hot vector corresponding to the special token $[M]$, and $\mathbf{Y_s[i]}$ is a one-hot vector corresponding to the token $\boldsymbol{y}_s^i$. It has been shown that this forward process has the following marginal distribution:

$$q(\boldsymbol{y}_t[i]|\boldsymbol{y}_0[i]) = \text{Cat}(\boldsymbol{y}_t[i]; (1-t)\mathbf{Y_0[i]} + t\mathbf{M}). \tag{7}$$

MDLM (Sahoo et al., 2024) shows that the posterior of the reverse process $p(\boldsymbol{y}_s|\boldsymbol{y}_t, \boldsymbol{y}_0)$ has the following form:

$$p(\boldsymbol{y}_s[i]|\boldsymbol{y}_t[i], \boldsymbol{y}_0[i]) = \begin{cases} \text{Cat}(\boldsymbol{y}_s[i]; \mathbf{Y_t}[i]), & \text{if } \boldsymbol{y}_s[i] \neq [M] \\ \text{Cat}(\boldsymbol{y}_s[i]; \frac{t-s}{t}\mathbf{Y_0}[i] + \frac{s}{t}\mathbf{M}), & \text{if } \boldsymbol{y}_s[i] = [M]. \end{cases} \tag{8}$$

In inference, the clean sequence $\boldsymbol{y}_0$ is not known at start, so it is replaced with the prediction from a policy network $\pi_\theta(\boldsymbol{y}_0[i]|\boldsymbol{y}_t)$, which gives the following empirical sampling process:

$$p_\theta(\boldsymbol{y}_s[i]|\boldsymbol{y}_t) = \begin{cases} \text{Cat}(X_s^i; \mathbf{Y_t}[i]), & \text{if } \boldsymbol{y}_s[i] \neq [M] \\ \text{Cat}(X_s^i; \frac{t-s}{t}\pi_\theta(\boldsymbol{y}_0[i]|\boldsymbol{y}_t) + \frac{s}{t}\mathbf{M}), & \text{if } X_s^i = [M]. \end{cases} \tag{9}$$

**SFT training process.** During SFT training, the following maximum likelihood estimation objective is adopted:

$$\mathcal{L}_{\text{SFT}} = -\mathbb{E}_{(\boldsymbol{y},\boldsymbol{x})\sim\mathcal{D}}[\log \pi_\theta(\boldsymbol{y}|\boldsymbol{x})] \tag{10}$$

where the likelihood is estimated via an MC estimator described in Sec. 3.3 of the main paper. We provide further discussion of estimating the likelihood in Appendix A.5.

## A.2. Unified paradigm for post-training

In this section, we provide a theoretical justification for unifying various post-training objectives into a weighted policy gradient method. We will include the derivation for GRPO (Guo et al., 2025), Online DPO and its variants (Guo et al., 2024b; Zhao et al., 2023), SFT, Best-of-N self-distillation (Sessa et al., 2024). We will show that these objectives share the same gradients as certain policy-gradient objectives, with a special weight.

**GRPO** We consider the setup of **fully-online** GRPO with strength of KL regularization $\beta = 0$. In this case, the GRPO objective is simplified to

$$J_{\text{grpo}}(\theta) = \mathop{\mathbb{E}}_{\boldsymbol{x}^i \sim \pi_{\text{old}}}\left[\frac{1}{N}\sum_{i=1}^N \min\left(\frac{\pi_\theta(\boldsymbol{y}^i|\boldsymbol{x})}{\pi_{\text{old}}(\boldsymbol{y}^i|\boldsymbol{x})}A_i^{\text{GRPO}}, \text{clip}\left(\frac{\pi_\theta(\boldsymbol{y}^i|\boldsymbol{x})}{\pi_{\text{old}}(\boldsymbol{y}^i|\boldsymbol{x})}, 1-\varepsilon, 1+\varepsilon\right)A_i^{\text{GRPO}}\right)\right]$$

In a pure on-policy setup, $\pi_{\text{old}}(\boldsymbol{y}^i|\boldsymbol{x}) = \text{sg}(\pi_\theta)(\boldsymbol{y}^i|\boldsymbol{x})$. Using the fact that $\nabla\pi_\theta(\boldsymbol{y}^i|\boldsymbol{x}) = \text{sg}(\pi_\theta)(\boldsymbol{y}^i|\boldsymbol{x})\nabla\log\pi_\theta(\boldsymbol{y}^i|\boldsymbol{x})$, we can express the gradient of the GRPO objective as,

$$\nabla J_{\text{grpo}}(\theta) = \mathop{\mathbb{E}}_{\boldsymbol{x}^i \sim \pi_\theta}\Big[\frac{1}{N}\sum_{i=1}^{N}\min\Big(\frac{\text{sg}(\pi_\theta(\boldsymbol{y}^i|\boldsymbol{x}))}{\text{sg}(\pi_\theta(\boldsymbol{y}^i|\boldsymbol{x}))}A_i^{\text{GRPO}}, \text{clip}\Big(\frac{\text{sg}(\pi_\theta(\boldsymbol{y}^i|\boldsymbol{x}))}{\text{sg}(\pi_\theta(\boldsymbol{y}^i|\boldsymbol{x}))}, 1-\varepsilon, 1+\varepsilon\Big)A_i^{\text{GRPO}}\Big)\nabla\log\pi_\theta(\boldsymbol{y}^i|\boldsymbol{x})\Big]$$

$$= \mathop{\mathbb{E}}_{\boldsymbol{x}^i \sim \pi_\theta}\Big[\frac{1}{N}\sum_{i=1}^{N}\min\Big(A_i^{\text{GRPO}}, \text{clip}\big(1, 1-\varepsilon, 1+\varepsilon\big)A_i^{\text{GRPO}}\Big)\nabla\log\pi_\theta(\boldsymbol{y}^i|\boldsymbol{x})\Big]$$

$$= \mathop{\mathbb{E}}_{\boldsymbol{x}^i \sim \pi_\theta}\Big[\frac{1}{N}\sum_{i=1}^{N}A_i^{\text{GRPO}}\nabla\log\pi_\theta(\boldsymbol{y}^i|\boldsymbol{x})\Big]$$

$$= \nabla\mathop{\mathbb{E}}_{\boldsymbol{x}^i \sim \pi_\theta}\Big[\frac{1}{N}\sum_{i=1}^{N}A_i^{\text{GRPO}}\log\pi_\theta(\boldsymbol{y}^i|\boldsymbol{x})\Big]$$

This is exactly policy gradient objectives with advantage $A_i = A_i^{\text{GRPO}}$

**Online DPO** For Online DPO (Guo et al., 2024b), and its variants, Online SiLC (Zhao et al., 2023), we generate the preference pairs from self-generated rollouts using the following protocol. After obtaining a group of responses $\boldsymbol{y}^1, \ldots, \boldsymbol{y}^N$, we re-order the responses so that the reward values $r_1, \ldots, r_N$ are monotonically decreasing as the response index grows. This is to say, we ensure $r_1 \geq r_2 \geq \cdots \geq r_N$. Then, we create a preference pair data by matching $\boldsymbol{y}^1$ with $\boldsymbol{y}^N$, $\boldsymbol{y}^2$ with $\boldsymbol{y}^{N-1}$, etc, where we consider $\boldsymbol{y}^1, \boldsymbol{y}^2$ as the positive data and $\boldsymbol{y}^{N-1}, \boldsymbol{y}^N$ as the negative data. Since the online DPO-type objectives are computed for a pair of positive and negative data points, we illustrate their derivation using the notation $\boldsymbol{y}^+, \boldsymbol{y}^-$. Note that the DPO objective is given as

$$J_{\text{DPO}}(\theta) = \mathbb{E}_{\boldsymbol{y}^+,\boldsymbol{y}^- \sim \pi_\theta(\cdot|\boldsymbol{x})}\Big[\log\sigma\big(z_\theta(\boldsymbol{x}, \boldsymbol{y}^+, \boldsymbol{y}^-)\big)\Big],$$

where $\sigma$ is the sigmoid function that is defined as $\sigma(x) = \frac{1}{1+e^{-x}}$, and it satsifies $\sigma(-x) = 1 - \sigma(x)$, and $z_\theta$ is define as

$$z_\theta(\boldsymbol{x}, \boldsymbol{y}^+, \boldsymbol{y}^-) = \beta\Big(\log\pi_\theta(\boldsymbol{y}^+ \mid \boldsymbol{x}) - \log\pi_\theta(\boldsymbol{y}^- \mid \boldsymbol{x}) - \log\pi_{\text{ref}}(\boldsymbol{y}^+ \mid \boldsymbol{x}) + \log\pi_{\text{ref}}(\boldsymbol{y}^- \mid \boldsymbol{x})\Big).$$

Therefore, computing the gradient of the DPO objective gives,

$$\nabla J_{\text{DPO}}(\theta) = \mathbb{E}_{\boldsymbol{y}^+,\boldsymbol{y}^- \sim \pi_\theta(\cdot|\boldsymbol{x})}\Big[\nabla\log\sigma(z_\theta)\Big]$$

$$= \mathbb{E}_{\boldsymbol{y}^+,\boldsymbol{y}^- \sim \pi_\theta(\cdot|\boldsymbol{x})}\Big[\frac{\partial}{\partial z_\theta}\log\sigma(z_\theta)\,\nabla z_\theta\Big]$$

$$= \mathbb{E}_{\boldsymbol{y}^+,\boldsymbol{y}^- \sim \pi_\theta(\cdot|\boldsymbol{x})}\Big[\big(1-\sigma(z_\theta)\big)\,\nabla z_\theta\Big]$$

$$= \mathbb{E}_{\boldsymbol{y}^+,\boldsymbol{y}^- \sim \pi_\theta(\cdot|\boldsymbol{x})}\Big[\big(\sigma(-z_\theta)\big)\,\beta\,\nabla\Big(\log\pi_\theta(\boldsymbol{y}^+ \mid \boldsymbol{x}) - \log\pi_\theta(\boldsymbol{y}^- \mid \boldsymbol{x})\Big)\Big]$$

$$= \mathbb{E}_{\boldsymbol{y}^+,\boldsymbol{y}^- \sim \pi_\theta(\cdot|\boldsymbol{x})}\Big[\beta\cdot\sigma(-z_\theta)\Big(\nabla\log\pi_\theta(\boldsymbol{y}^+ \mid \boldsymbol{x}) - \nabla\log\pi_\theta(\boldsymbol{y}^- \mid \boldsymbol{x})\Big)\Big]$$

$$= \mathbb{E}_{\boldsymbol{y}^+,\boldsymbol{y}^- \sim \pi_\theta(\cdot|\boldsymbol{x})}\Big[\beta\cdot\sigma(-z_\theta)\nabla\log\pi_\theta(\boldsymbol{y}^+|\boldsymbol{x}) + (-\beta\cdot\sigma(-z_\theta))\nabla\log\pi_\theta(\boldsymbol{y}^-|\boldsymbol{x})\Big]$$

$$= \nabla\mathbb{E}_{\boldsymbol{y}^+,\boldsymbol{y}^- \sim \pi_\theta(\cdot|\boldsymbol{x})}\Big[\beta\cdot\sigma(-z_\theta)\log\pi_\theta(\boldsymbol{y}^+|\boldsymbol{x}) + (-\beta\cdot\sigma(-z_\theta))\log\pi_\theta(\boldsymbol{y}^-|\boldsymbol{x})\Big]$$

This is the same as weighted policy gradient objectives with advantage $\beta\cdot\sigma(-z_\theta)$ assigned to $\boldsymbol{y}^+$ and $-\beta\cdot\sigma(-z_\theta)$ assigned to $\boldsymbol{y}^-$.

**Online DPO-smooth** We can also create a smoothed version of Online DPO to alleviate label noise arising from inaccurate preference pairs, which stem from the inherent flaws of the reward models. Let $\varepsilon$ be a label smooth/noise parameter, indicating that with probability $\varepsilon$, the obtained preference is wrong. Then, after taking this into consideration, the correct gradient for Online DPO-smooth should be

$$\nabla J_{\text{DPO-smooth}} = \nabla\mathbb{E}_{\boldsymbol{y}^+,\boldsymbol{y}^- \sim \pi_\theta(\cdot|\boldsymbol{x})}\Big[\beta\cdot\Big(\big((1-\varepsilon)\cdot\sigma(-z_\theta) - \varepsilon\sigma(z_\theta)\big)\log\pi_\theta(\boldsymbol{y}^+|\boldsymbol{x}) - \big((1-\varepsilon)\cdot\sigma(-z_\theta) - \varepsilon\sigma(z_\theta)\big)\log\pi_\theta(\boldsymbol{y}^-|\boldsymbol{x})\Big)\Big]$$

This is the same as weighted policy gradient objectives with advantage $(1 - \varepsilon) \cdot \sigma(-z_\theta) - \varepsilon \sigma(z_\theta)$ assigned to $\boldsymbol{y}^+$ and $-\big((1 - \varepsilon) \cdot \sigma(-z_\theta) - \varepsilon \sigma(z_\theta)\big)$ assigned to $\boldsymbol{y}^-$.

**Online SLiC**   Similarly, we can compute derive for SLiC (Zhao et al., 2023),

$$J_{\text{SLiC}}(\theta) := \mathbb{E}_{\boldsymbol{y}^+, \boldsymbol{y}^- \sim \pi_\theta(\cdot|\boldsymbol{x})} \Big[ - \max \big(0, \ \tau - \log \pi_\theta(\boldsymbol{y}^+ \mid \boldsymbol{x}) + \log \pi_\theta(\boldsymbol{y}^- \mid \boldsymbol{x})\big) \Big].$$

where $\tau$ is a pre-defined threshold value. Define the margin violation indicator

$$\mathbb{I}_{\text{viol}} = \mathbf{1} \big[ \tau - \log \pi_\theta(\boldsymbol{y}^+ \mid \boldsymbol{x}) + \log \pi_\theta(\boldsymbol{y}^- \mid \boldsymbol{x}) > 0 \big].$$

Therefore, the gradient of the objective

$$\nabla J_{\text{SLiC}}(\theta) = \mathbb{E}_{\boldsymbol{y}^+, \boldsymbol{y}^- \sim \pi_\theta(\cdot|\boldsymbol{x})} \Big[ \mathbb{I}_{\text{viol}} \nabla \Big( \log \pi_\theta(\boldsymbol{y}^+ \mid \boldsymbol{x}) - \log \pi_\theta(\boldsymbol{y}^- \mid \boldsymbol{x}) \Big) \Big]$$

$$= \mathbb{E}_{\boldsymbol{y}^+, \boldsymbol{y}^- \sim \pi_\theta(\cdot|\boldsymbol{x})} \Big[ \mathbb{I}_{\text{viol}} \Big( \nabla \log \pi_\theta(\boldsymbol{y}^+ \mid \boldsymbol{x}) - \nabla \log \pi_\theta(\boldsymbol{y}^- \mid \boldsymbol{x}) \Big) \Big].$$

This is the same as the gradient of the weighted policy gradient objective with advantage $\mathbb{I}_{\text{viol}}$ assigned to $\boldsymbol{y}^+$ and advantage $-\mathbb{I}_{\text{viol}}$ assigned to $\boldsymbol{y}^-$

**SFT and Best-of-N self-distillation**   It's straightforward to see that, for SFT, the gradient is the same as policy gradient with constant advantage value 1 across all samples as the objectives are equivalent. For Best-of-N self-distillation (Sessa et al., 2024), the objective is given as

$$\mathcal{L}_{\text{distill}}(\theta) = \text{KL}(\pi_{\text{BoN}} || \pi_\theta) = \mathbb{E}_{\boldsymbol{y}^i \sim \pi_{\text{BoN}}(\cdot|\boldsymbol{x})} \Big[ \log \frac{\pi_{\text{BoN}}(\boldsymbol{y}^i|\boldsymbol{x})}{\pi_\theta(\boldsymbol{y}^i|\boldsymbol{x})} \Big]$$

Therefore, we can write its gradient as,

$$\nabla \mathcal{L}_{\text{distill}}(\theta) = \nabla \mathbb{E}_{\boldsymbol{y}^i \sim \pi_{\text{BoN}}(\cdot|\boldsymbol{x})} \Big[ \log \frac{\pi_{\text{BoN}}(\boldsymbol{y}^i|\boldsymbol{x})}{\pi_\theta(\boldsymbol{y}^i|\boldsymbol{x})} \Big] = \mathbb{E}_{\boldsymbol{y}^i \sim \pi_{\text{BoN}}(\cdot|\boldsymbol{x})} \Big[ \nabla \log \pi_\theta(\boldsymbol{y}^i|\boldsymbol{x}) \Big]$$

Since $\boldsymbol{y}^i \sim \pi_{\text{BoN}}(\cdot|\boldsymbol{x})$ represents that $\boldsymbol{y}^i$ is the one with highest reward $r_i$ among $r_1, \ldots, r_N$, the objective can be simplified to weighted policy gradient objective with advantage 1 assigned to the best sequence with highest reward $\boldsymbol{y}^i$ and other wise 0. This is equivalent to performing SFT only on the self-generated best sequence.

### A.3. Answer Forcing

In this section, we provide a detailed account of the proposed answer-forcing algorithm. This technique is applicable to tasks with verifiable rewards, where the reward is computed by checking the generated answer against a ground truth, such as the 0-1 correctness reward for math problem and IoU reward for object grounding.

Given a group size of $N$, the naive implementation of answer-forcing described in Section 3.2.1 would first generate $N$ samples, evaluate the rewards, and then decide whether to generate an additional sample via injection. This is highly inefficient. Instead, we always generate $N + 1$ samples in parallel for each group, with 1 sample containing a ground-truth answer. However, depending on the rewards of the first $N$ samples, we optionally discard the extra sample from the loss computation when the remaining $N$ samples already include outputs with a significantly effective training signal (e.g., high rewards as measured by accuracy or IoU).

Concretely, during the online sampling process, we are given a prompt $x$, ground truth answer $z^*$ and a reward function $R$. When the desired group size is $N$, we always generate $N + 1$ samples with one extra answer-forced sample. Specifically, we initialize timestamps $t_1 = t_2 = \ldots t_N = 1$ and initialize $\boldsymbol{y}^1 = \boldsymbol{y}^2 .. \boldsymbol{y}^N$ using a fully masked sequence "M M . . .M". We initialize $\boldsymbol{y}_{t_{N+1}}^{N+1} =$ "MM . . . M <answer> $z^\star$ </answer>" with the answer section pre-filled. $t_{N+1}$ is set to the value $t'$ according to the mask ratio. For example, if the answer has 3 tokens and max generation length is 12, the timestep $t'$ will be 0.75, since $\frac{9}{12}$ of the tokens are masked.

---

**Algorithm 1** Answer Forcing with dLLMs

---

**Require:** Prompt $\boldsymbol{x}$, policy $\pi_\theta$, group size $N$, ground-truth answer $z^\star$, reward function $R$, threshold $\tau$, injection ration $\beta$
**Ensure:** Training group $\mathcal{G}$

 1: Initialize forced sample:
 2: **for** $i = 1$ to $N$ **do**
 3:    $\boldsymbol{y}_1^i \leftarrow$ "M M . . . M "
 4:    $t_i \leftarrow 1$
 5: **end for**
 6: $\boldsymbol{y}_{t'}^{N+1} \leftarrow$ "M M . . . M `<answer>` $z^\star$ `</answer>`"
 7: $t_{N+1} \leftarrow t'$
 8: **for** $i = 1$ to $N + 1$ **in parallel do**
 9:    $\boldsymbol{y}^i \sim \pi_\theta(\boldsymbol{y}_0^i \mid \boldsymbol{x}, \boldsymbol{y}_{t_i}^i)$
10: **end for**
11: **for** $i = 1$ to $N$ **do**
12:    $r^i \leftarrow R(\boldsymbol{x}, \boldsymbol{y}^i, z^*)$
13: **end for**
14: $r_{\max} \leftarrow \max_{i \in [N]} r^i$
15: **if** $r_{\max} < \tau$ and $Rand() < \beta$ **then**
16:    $\mathcal{G} \leftarrow \{(\boldsymbol{x}, \boldsymbol{y}^i)\}_{i=2}^{N+1}$
17: **else**
18:    $\mathcal{G} \leftarrow \{(\boldsymbol{x}, \boldsymbol{y}^i)\}_{i=1}^{N}$
19: **end if**
20: **return** $\mathcal{G}_{\text{loss}}$

---

All of these sequences have equivalent length, which is set to 512 for math reasoning, 128 for object grounding and 256 for image editing based on the distribution of reasoning lengths in our SFT data. Ideally, we do not want the generated sequences to have exactly 512, 256, or 128 tokens to allow for some flexibility. In standard sampling, the model generates special [PAD] tokens at the end of the sentence if the reasoning length is less than the maximum sequence length. For the answer-forced sample, we adopted the Fill-in-the-Middle (FIM) design of LaViDa (Li et al., 2025c) and inserted random-length "[S] . . . [S]" sequences in the SFT data right before "`<answer>`. . .`</answer>`", where [S] is a special infilling token. This design allows flexible-length text infilling as the model can generate "[S] . . . [S]" if the reasoning length is less than the lengths of mask segment "[M] . . . [M]" in $\boldsymbol{y}^{N+1}$.

After obtaining $\boldsymbol{y}_{t_1}^1 \ldots \boldsymbol{y}_{t_{N+1}}^{N+1}$, we perform diffusion sampling using the policy model $\pi_\theta$ to obtain $\boldsymbol{y}^i \sim \pi_\theta(\boldsymbol{y}_0^i \mid \boldsymbol{x}, \boldsymbol{y}_{t_i}^i)$ for $i = 1, 2 \ldots N, N + 1$. Notably, the model forward computation of these samples can be performed in parallel. We then evaluate the rewards on these $N + 1$ samples to obtain $r_1 \ldots r_{N+1}$. Finally, we check if the maximum rewards $r_{\max}$ among non-answer-forced samples exceeds a threshold $\tau$. In our setup, $\tau$ is set to 0.5 for both 0-1 math rewards and IoU rewards. We do not include any auxiliary rewards such as format rewards in this step. If $r_{\max}$ exceeds $\tau$, we consider the original sample to have high-quality outputs and discard the answer-forced sample. Otherwise, we randomly replace one samples in $\boldsymbol{y}^1 \ldots \boldsymbol{y}^N$ with $\boldsymbol{y}^{N+1}$. Since $\boldsymbol{y}^1 \ldots \boldsymbol{y}^N$ are just i.i.d samples, we always discard $\boldsymbol{y}^1$ in our implementation. The answer-forcing algorithm is formally documented in Algorithm 1.

### A.4. Tree Search

In this section, we provide detailed descriptions of our tree-search algorithm. This technique is applicable to tasks without ground-truth answers but with a real-valued reward function. It is not applicable to 0-1 rewards, since we cannot meaningfully identify a best sample when all rewards are zero.

Given a prompt $x$ and reward function $R$, and a base group size $N$, the tree search process is controlled by the number of tree expansions $k$ and max diffusion steps $T$, and restart timestep index $s_1 \ldots s_k \in \{0, 1..T\}$. In particular, the restart timestep index determines which point in the saved trajectories should serve as the branching point. $s_1$ is always 0 since we always need to go through the full $T$ diffusion steps for the first $N$ samples $\boldsymbol{y}^1 \ldots \boldsymbol{y}^N$. The indices $s_1 \ldots s_k \in \{0, 1..T\}$ directly correspond to diffusion timesteps $t_1 \ldots t_N \in [0, 1]$ through the relation $t_i = 1 - \frac{s_i}{T}$.

---

**Algorithm 2** Tree Search with dLLMs

---

**Require:** Prompt $\boldsymbol{x}$, policy $\pi_\theta$, reward function $R$, base group size $N$, number of tree expansions $k$, diffusion steps $T$, restart timestep index $s_1...s_k \in \{0, 1, ...T\}$ with $s_1 = 0$
**Ensure:** Generated group $\mathcal{G}$ of size $Nk$

 1: Initialize empty group $\mathcal{G} \leftarrow \emptyset$
 2: **for** $i = 1$ to $K$ **do**
 3:     $t_i \leftarrow 1 - \frac{s_i}{T}$                                             $\triangleright$ We always have $t_1 = 1 - \frac{s_1}{T} = 1$
 4: **end for**
 5: **for** $j = 1$ to $N$ **do**
 6:     $\boldsymbol{y}_{t_1}^j \leftarrow$ "MM ... M"                                         $\triangleright$ Fully masked sequence
 7: **end for**
 8: **for** $i = 1$ to $k$ **do**
 9:     **for** $j = Ni - N + 1$ to $Ni$ **in parallel do**
10:         $\boldsymbol{y}_0^j \sim \pi_\theta(\boldsymbol{y}_0 \mid \boldsymbol{x}, \boldsymbol{y}_{t_i}^j)$
11:         $r^j \leftarrow R(\boldsymbol{x}, \boldsymbol{y}_0^j)$
12:         Record unmasking order $v^j \in \{1, \ldots, T\}^L$
13:     **end for**
14:     Gather $\{r^j, v^j, \boldsymbol{y}^j\}_{j=Ni-N+1, Ni-N+2,...Ni}$ across processes
15:     $\mathcal{G} \leftarrow \mathcal{G} \cup \{(\boldsymbol{x}, \boldsymbol{y}_0^j)\}_{j=Ni-N+1, Ni-N+2,...Ni}$             $\triangleright$ $\mathcal{G}$ has $Ni$ samples after this point.
16:     $m \leftarrow \arg\max_{j\in\{1,2..Ni\}} r^j$
17:     $\boldsymbol{y}_{t_{i+1}}^m[\ell] \leftarrow \boldsymbol{y}_0^m[\ell]$ **if** $v^m[\ell] \leq s_{i+1}$, **otherwise** M,    $\forall \ell = 1, \ldots, L$     $\triangleright$ Recover restart state
18:     **for** $j = Ni + 1$ to $Ni + N$ **do**
19:         $\boldsymbol{y}_{t_{i+1}}^j \leftarrow \boldsymbol{y}_{t_{i+1}}^m$               $\triangleright$ Prepare next batch for restarting from partial state
20:     **end for**
21: **end for**
22: **return** $\mathcal{G}$                                      $\triangleright$ $\mathcal{G}$ has $Nk$ samples.

---

In particular, $t_1 = 1$ always holds, indicating fully masked sequences.

After the $i$th batch is generated, we will have $Ni$ samples. We find the index $m$ corresponding to the sample with the highest reward among all previously generated samples $\boldsymbol{y}^1.\boldsymbol{y}^{Ni}$. In the $(i+1)$th batch, we generate the $(Ni + 1)$th sample to the $(Ni + N)$th sample using $\boldsymbol{y}_{t_{i+1}}^m$ as the starting point. These samples will go through $T - s_{i+1}$ diffusion steps. This process is repeated until all $Nk$ samples are generated.

In a distributed training setup, samples in each batch are generated on multiple GPUs in parallel. Hence, we need to gather the generated trajectories and evaluated rewards. To reduce the cost of maintaining and synchronizing multiple trajectories in memory, we use a more compact representation, leveraging the fact that once a token is unmasked at a diffusion step, it will not be modified in subsequent steps.

Concretely, we store the final generated result $\boldsymbol{y}_0^j$ and an array $v^j \sim \{1, ..T\}^L$ which keeps track of when each token is unmasked, where $L$ is the sequence length. $v^j[i] = p$ indicates the i-$th$ token is unmasked at p-$th$ diffusion step where $1 \leq p \leq T$. To recover $\boldsymbol{y}_{t'}^j$ for arbitrary $t' \in [0, 1]$, we can obtain the corresponding diffusion step index $s'$ through the relationship $t_i = 1 - \frac{s_i}{T}$ and recover $\boldsymbol{y}_{t'}^j$ through $\boldsymbol{y}_{t'}^j[i] = \boldsymbol{y}_0^j[i]$ if $v^j[i] \leq s'$ and $\boldsymbol{y}_{t'}^j[i] = M$ otherwise. This reduces the overhead from $\mathcal{O}(NLT)$ to $\mathcal{O}(NL)$ at each batch. The tree search algorithm is formally described in Algorithm 2.

### A.5. Likelihood Estimator

In this section, we discuss more technical details of likelihood estimation in dLLMs. Recall that we estimate the data log-likelihood using the ELBO, as follows.

$$\log \pi_\theta(\boldsymbol{y}|\boldsymbol{x}) = \mathbb{E}_{t, \boldsymbol{y}_t}\left[w(t) \sum_{k\in\{k|\boldsymbol{y}_t[k]=\text{M}\}} \log \pi_\theta(\boldsymbol{y}[k]|\boldsymbol{y}_t, \boldsymbol{x})\right]$$

We will list several ELBO estimation methods from the literature for comparison with ours. A detailed visualization is presented in Figure 6.

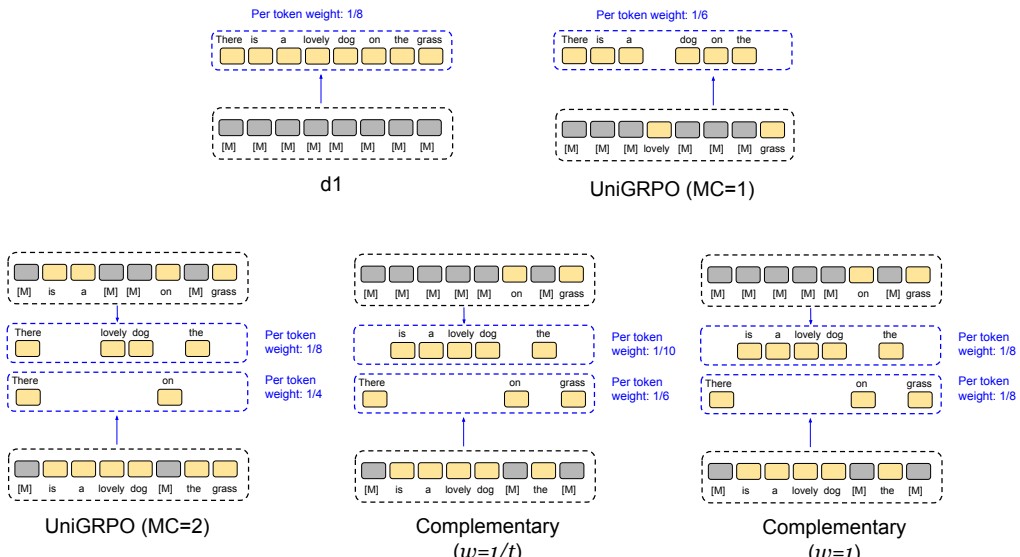

*Figure 6.* **Different Design Choices of Likelihood Estimators.** a) d1 uses MC=1, $w(t) = \frac{1}{t} = 1$ and always sample $t = 1$, equivalent to fully masked sequences. Hence, it can compute per-token likelihood at all positions. b) UniGPRO uses MC=1 and $w(t) = \frac{1}{t}$. It randomly masks a subset of tokens and computes the likelihood with only masked tokens. c) We can improve UniGRPO by increasing the MC to 2 for a broader token coverage. However, it leads to imbalanced gradients with $w(t) = \frac{1}{t}$. d) In vanilla complementary masking developed for pretraining, two MC samples are coupled to ensure 100% token coverage. However, the token imbalance issue persists because it uses $w(t) = \frac{1}{t}$. e) In our design, we set $w(t) = 1$, ensuring all-token coverage while balancing the importance of each token in a uniform fashion.

**d1 ([Zhao et al., 2025a](#))**  d1 estimates the likelihood by constantly sampling $t = 1$ and $\boldsymbol{y}_1 = [\mathrm{M}, \dots, \mathrm{M}]$, uses only one sample (MC = 1), with weight $w(t) = \frac{1}{t}$. This gives the estimation expression as,

$$\log \pi_\theta^{\mathrm{d1}}(\boldsymbol{y}|\boldsymbol{x}) = \sum_{k=1}^{L} \log \pi_\theta(\boldsymbol{y}[k]|\boldsymbol{y}_1, \boldsymbol{x})$$

**UniGRPO ([Yang et al., 2025b](#))**  UniGRPO instead sample mask ratio $t$ from $[0, 1]$ uniformly and a corresponding binary mask M associated with $\boldsymbol{y}_t$ based on such mask ratio (MC = 1). It adopts the weight $w(t) = \frac{1}{t}$. This means that it computes the data loglikelihood with,

$$\log \pi_\theta^{\mathrm{unigrpo}}(\boldsymbol{y}|\boldsymbol{x}) = \frac{1}{t} \sum_{k \in \{k|\boldsymbol{y}_t[k]=\mathrm{M}\}} -\log \pi_\theta(\boldsymbol{y}[k]|\boldsymbol{y}_t, \boldsymbol{x})$$

We could improve the design by increasing the Monte Carlo sample size to 2 (MC=2) by sampling another masked sample with a different mask rate, chosen also uniformly at random.

$$\log \pi_\theta^{\mathrm{unigrpo},2}(\boldsymbol{y}|\boldsymbol{x}) = \frac{1}{2}\Big[\frac{1}{t_1} \sum_{k \in \{k|\boldsymbol{y}_{t_1}[k]=\mathrm{M}\}} \log \pi_\theta(\boldsymbol{y}[k]|\boldsymbol{y}_{t_1}, \boldsymbol{x}) + \frac{1}{t_2} \sum_{k \in \{k|\boldsymbol{y}_{t_2}[k]=\mathrm{M}\}} \log \pi_\theta(\boldsymbol{y}[k]|\boldsymbol{y}_{t_2}, \boldsymbol{x})\Big]$$

**Our recipe**  We adopt a simple, effective approach for estimating the log probability using 2 samples with complementary masking (MC=2). We sample $\boldsymbol{y}_{t_1} \sim q(y_{t_1}|y_0)$ using the discrete forward diffusion process, and set

$$\boldsymbol{y}_{t_2}[i] = \begin{cases} \boldsymbol{y}[i], & \text{if } \boldsymbol{y}_{t_1}[i] = M, \\ M, & \text{if } \boldsymbol{y}_{t_1}[i] \neq M. \end{cases} \tag{11}$$

Finally, we compute the likelihood via

$$\log \pi_\theta^{\text{ours}}(\boldsymbol{y}|\boldsymbol{x}) = \frac{1}{2}\Bigg[ \sum_{k \in \{k|\boldsymbol{y}_{t_1}[k]=\text{M}\}} \log \pi_\theta(\boldsymbol{y}[k]|\boldsymbol{y}_{t_1},\boldsymbol{x}) + \sum_{k \in \{k|\boldsymbol{y}_{t_2}[k]=\text{M}\}} \log \pi_\theta(\boldsymbol{y}[k]|\boldsymbol{y}_{t_2},\boldsymbol{x}) \Bigg]$$

This approach differs from vanilla complementary masking used by a few other works (Li et al., 2025c; Zhu et al., 2025; Bie et al., 2025) in that we adopt $w(t) = 1$ as opposed to $w(t) = \frac{1}{t}$. While this seems like a small modification, it has a profound impact on model performance, as confirmed by the ablation study in B.3.

We visualize these likelihood estimators in Figure 6.

## B. Additional Experiment Details and Results

### B.1. Setup

**Training Dataset**

Our training consists of two stages. Stage 1 consists of only SFT objective. Stage 2 incorporates both online sampling and offline datasets in a unified fashion.

Stage 1 datasets can be generally categorized into two categories. The first category is a subset of LaViDa-O's pretraining data, which includes:

- *A: Text-to-Image Pairs.* We source data from LAION-2B (Schuhmann et al., 2022), COYO-700M (Byeon et al., 2022), BLIP3o-60k (Chen et al., 2025c), ShareGPT4o-Image (Chen et al., 2025b). Each dataset is heavily filtered to remove NSFW prompts, low CLIP scores (Radford et al., 2021), low aesthetic scores (Schuhmann, 2022), and low-resolution images following LaViDa-O's pipeline. We include all data from BLIP3o-60k and ShareGPT4o-Image, and select highest-quality samples from LAION and COYO based on CLIP scores and aesthetic scores. This part consists of 20M images.

- *B: Image-level Understanding Data.* We include MAmmoth-VL (Guo et al., 2024a), and VisualWebInstruct (Jia et al., 2025).

- *C: Region-level Understanding Data.* We include GranD (Rasheed et al., 2024) and RefCOCO (Kazemzadeh et al., 2014).

- *D: Image Editing Data.* We include ShareGPT4o-Image (Chen et al., 2025b), GPT-Edit-1.5M (Wang et al., 2025f), and the image editing subset of UniWorld-V1 (Hu et al., 2022).

The second category includes newly incorporated reasoning data not used in LaViDa-O's original training recipe. It includes

- *E: Visual Understanding with Reasoning.* We include Vision-R1-Cold(Huang et al., 2025), which consists of visual math problems and VQA problems.

- *F: Pure Language-based Reasoning.* We include DeepScalar (Luo et al., 2025a).

- *G: Image Editing with Reasoning.* We include GoT (Fang et al., 2025) data. Additionally, we run the data pipeline of GoT on GPT-Edit-1.5M (Wang et al., 2025f) to further expand the data.

- *H: Reason-intensive Grounding.* We include ReasonSeg (Lai et al., 2024) and Lisa-CoT (Yang et al., 2023).

Given the substantial imbalance between the two categories, we design a dataloader to prioritize the acquisition of reasoning capabilities. The dataset of the two categories is sampled with a 3:7 ratio.

Stage 2 training includes all datasets from Stage 1. It also incorporates additional RL datasets, including

- *I: Visual Understanding with Reasoning.* We include Vision-R1-Cold and Vision-R1-RL (Huang et al., 2025), which consists of visual math problems and VQA problems, and ViRL-39k (Wang et al., 2025b)

- *J: Language Based Reasoning.* We include DeepScalar (Luo et al., 2025a), GSM8K (Cobbe et al., 2021), MATH (Lightman et al., 2023).

- *K: Image Editing with Reasoning.* We include input images and prompts from EditScore-RL (Luo et al., 2025b).

- *L: Reason-intensive Grounding.* We include ReasonSeg (Lai et al., 2024), Lisa-CoT (Yang et al., 2023), RefCOCO (Kazemzadeh et al., 2014).

While some datasets such as DeepScalar (Luo et al., 2025a) appear both in the SFT data and RL data, they are incorporated in different manners. In RL data, the reasoning traces provided in these datasets are not used. Only the prompts and corresponding answers are used.

**Reward Functions** For math problems and question answering, we use 0-1 correctness reward. For object grounding, we use IoU reward. For image editing, we use a VLM-based reward model, Editscore(Luo et al., 2025b). Since Editscore requires considerable memory, it is hosted on a separate server to the main training processes. In addition to these key rewards, we also incorporate auxiliary rewards such as tag-formatting rewards and a repetition penalty, following existing work (Gao et al., 2024; Zhao et al., 2025a). We note that the formatting reward in existing works like d1 is not robust as it merely counts the "<think>" and "<answer>" tokens. We replace it with a more robust formatting reward based on regular expressions.

**Training Hyperparameter**

We report the training hyperparameters, including the learning rate, number of training steps, optimizer setup, and image resolution for understanding and generation tasks in Table 9 for reference. The training is conducted of 64 GPUs. The main experiment is conducted on H100s. Some ablation studies use A100s. The total training time takes 5 days for Stage 1 and 3 days for Stage 2. We observe a considerable bottleneck in reward evaluation. In particular, with a global batch size of 256 images, the edit score evaluation takes 70-140 seconds. We will investigate how to further optimize this infrastructure bottleneck in future work by improving the network infrastructure and scaling reward servers.

*Table 9.* **Training configurations of LaViDa-R1.** We report the relevant hyperparameters for training, including the learning rate, number of training steps, optimizer setup, image resolution for understanding and generation tasks.

|  | Stage 1 | Stage 2 |
|---|---|---|
| Learning Rate | $5 \times 10^{-6}$ | $5 \times 10^{-7}$ |
| Steps | 100k | 5k |
| $\beta_1$ | 0.99 | 0.99 |
| $\beta_2$ | 0.999 | 0.999 |
| optimizer | AdamW | AdamW |
| Dataset Used | A,B,C,D,E,F,G,H | A,B,C,D,E,F,G,H,I,J,K,L |
| Loaded Parameters | 10.4B | 10.4B |
| Trainable Parameters | 10.4B | 10.4B |
| Und. resolution | $384 \times \{(1,3),(2,2)\}$ | $384 \times \{(1,3),(2,2)\}$ |
| Gen. resolution | 1024 | 1024 |
| Loss | SFT | Unifed |

**Evaluation Protocol.** We use a max sequence length of 128 for object grounding, 256 for image editing and 512 for all other tasks to match the training setup. All other sampling parameters remain unchanged following LaViDa-O's inference protocol.

**B.2. Additional Ablation Studies**

In this section, we report additional ablation study results to verify the design of LaViDa-R1.

**B.3. Timestep Weighting**

In Table 7, we report results of varying the timestep weighting discussed in Section A.5. Specifically, we compare results of complementary masking with $w = \frac{1}{t}$ and $w = 1$, results show that $w = 1$ works better. Specifically, we observe that

Table 10. **Ablation Studies of Per-token Likelihood Weighting**

| Weighting | Lisa-Grounding | ImgEdit |
|---|---|---|
| LaViDa-O + SFT | 40.3 | 3.81 |
| $w(t) = \frac{1}{t}$ | 64.3 | 3.82 |
| $w(t) = 1$ | **65.0** | **3.90** |

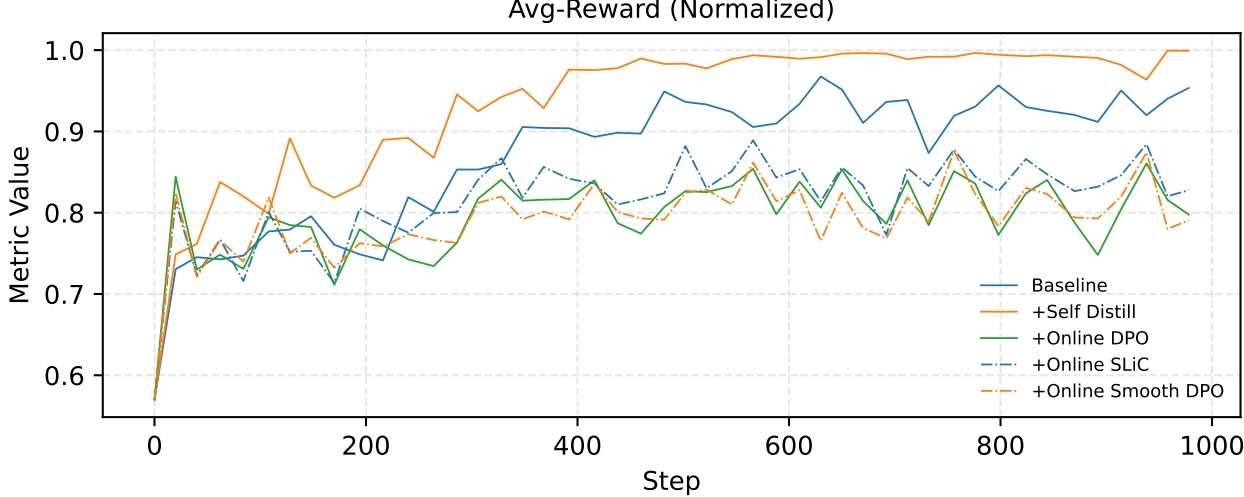

Figure 7. **Ablation Studies of Combining Different Losses in the Unified Framework**

$w = \frac{1}{t}$ significantly degrades performance on image-editing tasks. This may be attributed to the large number of tokens in visual generation tasks, which exacerbates the imbalance in per-token loss. In particular, image editing tasks produce 256 text tokens and 4096 image tokens per sample, significantly more than understanding-only tasks, which generate only 512 tokens.

### B.4. Alternative Losses in the Unified Framework

### B.5. Qualitative Results

In Figure 7, we visualize the reward trajectories of combining RL loss with other post-training methods beyond best-of-N distillation discussed in Section A.2. Specifically, we start with the GRPO+SFT baseline and explore incorporating additional losses, including online DPO, online DPO-smooth, and online SLiC loss. Notably, all these can be achieved by simply modifying the weight $A_i$ in Equation 5 without considerably changing the training pipeline. Results show that adding other losses is worse than the GRPO+SFT baseline, while incorporating Best-of-N distillation improves upon the baseline. Although most auxiliary losses did not improve performance, this experiment highlighted the flexibility of our proposed framework. We hope future work will further explore loss-weighting recipes beyond group advantages in GRPO.

To further demonstrate the effectiveness of LaViDa-R1, we include qualitative results of model outputs on object grounding tasks in Figure 8, image editing tasks in Figure 9 and math reasoning tasks in Figure 10. These results demonstrated the strong reasoning capabilities of LaViDa-R1 on diverse multimodal tasks.

## C. Limitation

Despite the strong results of LaViDa-R1, it has several key limitations. First, while LaViDa-R1 improves upon the base model LaViDa-O. There still exists a considerable gap between the reasoning performance of multimodal dLLMs and state-of-the-art AR MLLMs such as Qwen3-VL (Bai et al., 2025a). We hope that future work on scaling and improving pretraining can close this gap for base models. Second, while AR LLMs can leverage many efficient inference frameworks

such as vLLM (Kwon et al., 2023), these low-level optimization frameworks have not yet been fully adapted to support dLLMs. This results in a throughput bottleneck because our training process uses eager Python execution during online sampling. We hope future acceleration frameworks tailored to dLLMs will help address this issue. Lastly, while LaViDa-R1 already included diverse number of tasks and significantly expanded the scope of dLLM RL literature, there are more tasks to explore for unified multimodal dLLMs, such as multi-turn website code generation with visual feedback, interleaved text-image reasoning, poster design, etc. In this work, we focused on single-round reasoning with visual inputs. We will explore more tasks in future work.

**Text-to-Image Generation**.We also briefly explored extending reasoning to text-to-image (T2I) generation. However, we find that existing reward models are fundamentally misaligned with reasoning-centric objectives. Despite this limitation, LaViDa-R1 demonstrates non-trivial zero-shot reasoning ability on T2I tasks. An illustrative example is shown in Figure 11: given the prompt "The light source that replaced candles in homes during the early 20th century," LaViDa-R1 correctly reasons about the historical context and generates light bulbs, whereas LaViDa-O produces images of candles.

Unfortunately, current reward models fail to reliably distinguish such reasoning-grounded generations. Early approaches such as PickScore (Kirstain et al., 2023), which are built upon CLIP-based models, lack explicit reasoning capabilities. We further investigated more recent VLM-based reward models, but even the state-of-the-art UnifiedReward-Qwen-7B (Wang et al., 2025g) is unable to correctly rank the samples in this example. Upon inspecting the model's justifications, we find that it hallucinates spurious criteria, assessing alignment with surface-level concepts such as "candles" (object) and "replaced" (activity), thereby misinterpreting the compositional and historical reasoning required by the prompt.

Recent work (Wei et al., 2025) proposes using a frontier VLM (GPT-4.1) as an online reward model, which may alleviate some of these issues. However, this approach is prohibitively expensive and difficult to scale. We believe that effective and scalable reward modeling remains a key open challenge for reasoning-driven T2I generation. In light of these limitations, we leave reinforcement learning fine-tuning for T2I reasoning to future work.

# D. Additional Discussions for Related Works

**Reinforcement Learning with dLLMs.** d1 (Zhao et al., 2025a) first explored adapting the GPRO algorithm through token log probability ratios estimated through one-sample ELBO, with many works exploring other improved designs of estimation (Gong et al., 2025; Yang et al., 2025b; Wang et al., 2025a). In another line of work, wd1 (Tang et al., 2025) and DMPO (Zhu et al., 2025) addressed the dLLM RL tasks from a policy distribution-matching perspective, leading to new ways of computing advantage that differ from the aforementioned GRPO-style algorithms. TraceRL (Wang et al., 2025e) considers an actor-critic style method, and ESPO (Ou et al., 2025) adapts GSPO (Zheng et al., 2025a) to dLLMs.

**Unified SFT and RL.** There have been multiple works in the LLM community that explored effective strategies to combine SFT and RL into a single stage (Chen et al., 2025a; Lv et al., 2025; Liu et al., 2025b; Fu et al., 2025; Yan et al., 2025; Ma et al., 2025). These works have introduced approaches that dynamically balance off-policy SFT loss and on-policy RL loss, attempting to outperform single-stage RL or single-stage SFT for LLM post-training. Unlike these approaches, our loss formulation for unified SFT and RL is simple and was introduced to ensure the stability of the RL stage, rather than to make single-stage RL or single-stage SFT obsolete. Moreover, to the best of our knowledge, our work is the first to apply a unified SFT-RL strategy to unified multimodal models, whereas prior work has largely focused on single-text modalities using LLMs.

**Connection between Answer Forcing and IGPO.** IGPO (Zhao et al., 2025b) injects partially masked ground-truth reasoning traces as a hint in the online sampling process of dLLMs. While this design and our proposed answer forcing algorithm both make use of inpainting capabilities of dLLMs, they are distinct from each other in that IGPO requires high-quality ground-truth reasoning traces, whereas Answer Forcing only requires access to the final answer and does not require any pre-existing reasoning traces. In general, high-quality reasoning traces are rare and difficult to obtain for tasks such as object grounding, making answer forcing a more generalizable approach.

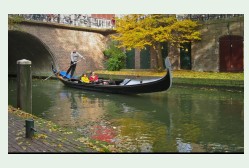 Who in the picture would be the most experienced in taking small boats to tour the sights? Please provide the bounding box coordinate of this region.

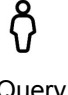
Query

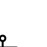
LaViDa-R1

<think> The person in the boat is likely the one steering the boat, as they are operating the boat and have experience in navigating boats.
- This individual is the person steering the boat.</think>
<answer> [256,360,324,480] </answer>

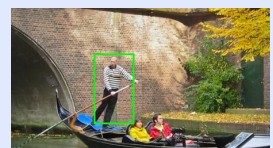

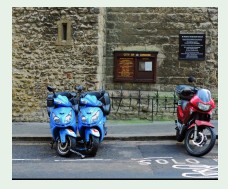 Please provide the bounding box coordinate of the motorcycles tied together.

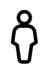
Query

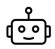
LaViDa-R1

<think> The motorcycles tied together are the two blue ones.
- The blue motorcycles are located on the left side of the image.
- The blue motorcycles are identical. They are tied together.</think>
<answer>[156,467,488,840]</answer>

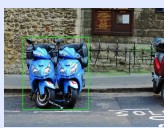

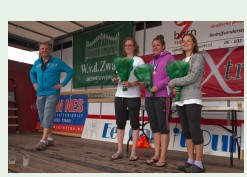 Please provide the bounding box coordinate of the person who is not likely to be a player

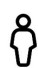
Query

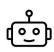
LaViDa-R1

<think>- The person on the far left of the image, wearing a blue jacket and standing casually, is not likely to be a player. This person is the only one not likely to be a player.</think>
<answer> [92,264,292,748] </answer>

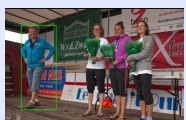

*Figure 8.* **Qualitative Results of Reason-Intensive Object Grounding.**

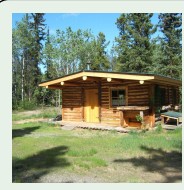 Change the cabin environment from a forest to a tropical beach setting.

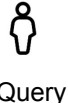 Query

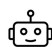 LaViDa-R1

<think>This edit requires modifying the whole image. The source image shows a log cabin in a forest setting with tall trees and green grass. The cabin and its immediate surroundings need to be transformed from a forest to a tropical beach environment. The background should be replaced with a sandy beach, turquoise ocean, palm trees, and a clear blue sky. The ground should be changed from grass to sand, and the trees should be replaced with palm trees. The cabin remains unchanged in structure and appearance, but the surrounding environment is completely altered.</think>
<answer>[img]</answer>

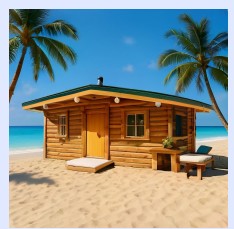

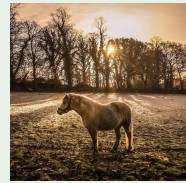 Transfer the image into a stained-glass cathedral-window style.

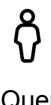 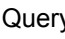 Query

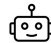 LaViDa-R1

<think> This edit requires modifying the whole image. The source image shows a horse standing in a field with trees in the background under a sunset sky. The entire image needs to be transformed into a stained-glass style, with bold outlines and vibrant colors. The horse, trees, and sky should be reimagined with geometric shapes and colored blocks, mimicking the appearance of stained glass. After the edit, the image will appear as a stained-glass window, with the horse, trees, and sky rendered in a stylized, colorful manner. </think>
<answer>[img]</answer>

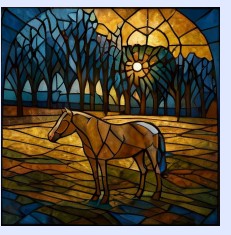

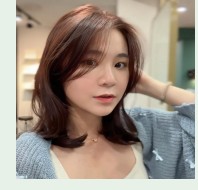 Upgrade the necklace's material to 999 pure gold.

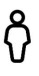 Query

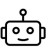 LaViDa-R1

<think>  This edit requires modifying only the specific regions. The source image shows a woman wearing a necklace with a simple design. The object to be edited is the necklace. The specific area to be edited is the necklace around the woman's neck. The specific change is to upgrade the necklace's material to 999 pure gold, making it appear more luxurious and reflective. After the edit, the necklace will appear as a shiny, polished gold piece, enhancing the overall aesthetic of the image. </think>
<answer>[img]</answer>

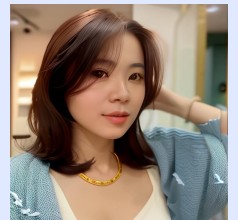

*Figure 9.* **Qualitative Results of Reason-Based Image Editing.**

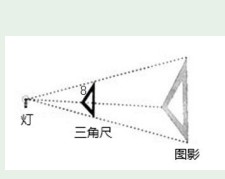 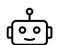 As shown in the figure, if the ratio of the distance from the bulb to the vertex of the triangle ruler to the distance from the bulb to the corresponding vertex of the triangular ruler projection is 2.0:5.0, Then the corresponding edge length of the projection triangle is
Choices:
A:8cm  B:20cm C:3.2cm D:10cm

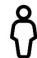
Query

 LaViDa-R1

<think> Okay, let's see. The problem illustrates that EF is parallel to BC, and AC bisects angle BAF. We need to find the measure of angle C. First, since EF is parallel to BC, and AB is a transversal, the corresponding angles should be equal. That means angle EAC is equal to angle ABC. But wait, angle EAC is part of angle BAF, right? Because AC bisects angle BAF. So, angle BAF is split into two equal parts by AC. If angle BAF is 130 degrees, then each part would be 65 degrees. Therefore, angle EAC is 65 degrees. But angle EAC is equal to angle ABC because EF is parallel to BC. So angle ABC is also 65 degrees. Now, in triangle ABC, we know angle ABC is 65 degrees and angle B is 50 degrees. So angle C would be 180 - 65 - 50 = 65 degrees. But wait, angle C is 65 degrees? That seems right. Let me check again. Wait, maybe I made a mistake. Let me think again. Since EF is parallel to BC, angle EAC is equal to angle ABC. So angle EAC is 65 degrees, so angle ABC is 65 degrees. Then in triangle ABC, angle B is 50 degrees, so angle C is 180 - 65 - 50 = 65 degrees. Yeah, that's correct. So the answer should be D. </think>
<answer> **D** </answer>

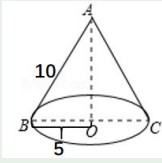 As shown in the picture, and the lateral area of the cone-shaped tent roof (excluding the seams) is
Choices:
A:15πm^2 B:30πm^2 C:50πm^2 D:75πm^2

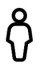
Query

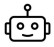 LaViDa-R1

<think> Okay, let's see. The problem is about finding the lateral (or curved) surface area of a cone-shaped tent. The given information includes the radius of the base (5 meters) and the slant height (10 meters). The formula for the lateral surface area of a cone is πrl, where r is the radius and l is the slant height. So, plugging in the values, it should be π * 5 * 10. Let me calculate that. 5 times 10 is 50, so 50π. Looking at the options, option C is 50π. That must be the correct answer. </think>
<answer> **C** </answer>

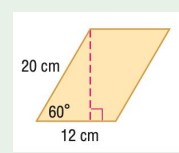 Find the perimeter of the parallelogram. Round to the nearest tenth if necessary.
Choices:
A:60  B:62  C:64 D:66

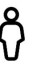
Query

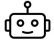 LaViDa-R1

<think> Okay, let's see. The problem is about finding the perimeter of a parallelogram. The given information includes two sides: 20 cm and 12 cm, and an angle of 60 degrees. First, I remember that in a parallelogram, opposite sides are equal. So, if one side is 20 cm, the opposite side must also be 20 cm. Similarly, the other side is 12 cm, so its opposite is also 12 cm. The perimeter of a parallelogram is calculated by adding up all the sides. Since opposite sides are equal, I can use the formula: Perimeter = 2 × (side1 + side2). Plugging in the values, that would be 2 × (20 + 12) = 2 × 32 = 64 cm. Wait, but the angle is 60 degrees. Does that affect the perimeter? No, the angle doesn't change the lengths of the sides. The perimeter is purely based on the lengths of the sides, which are given as 20 cm and 12 cm. So the perimeter should be 64 cm. Let me double-check. If the sides are 20 cm and 12 cm, then 20 + 12 is 32, and multiplying by 2 gives 64. Yep, that's correct. </think>
<answer> **C** </answer>

*Figure 10.* **Qualitative Results on Visual Math Problems.**

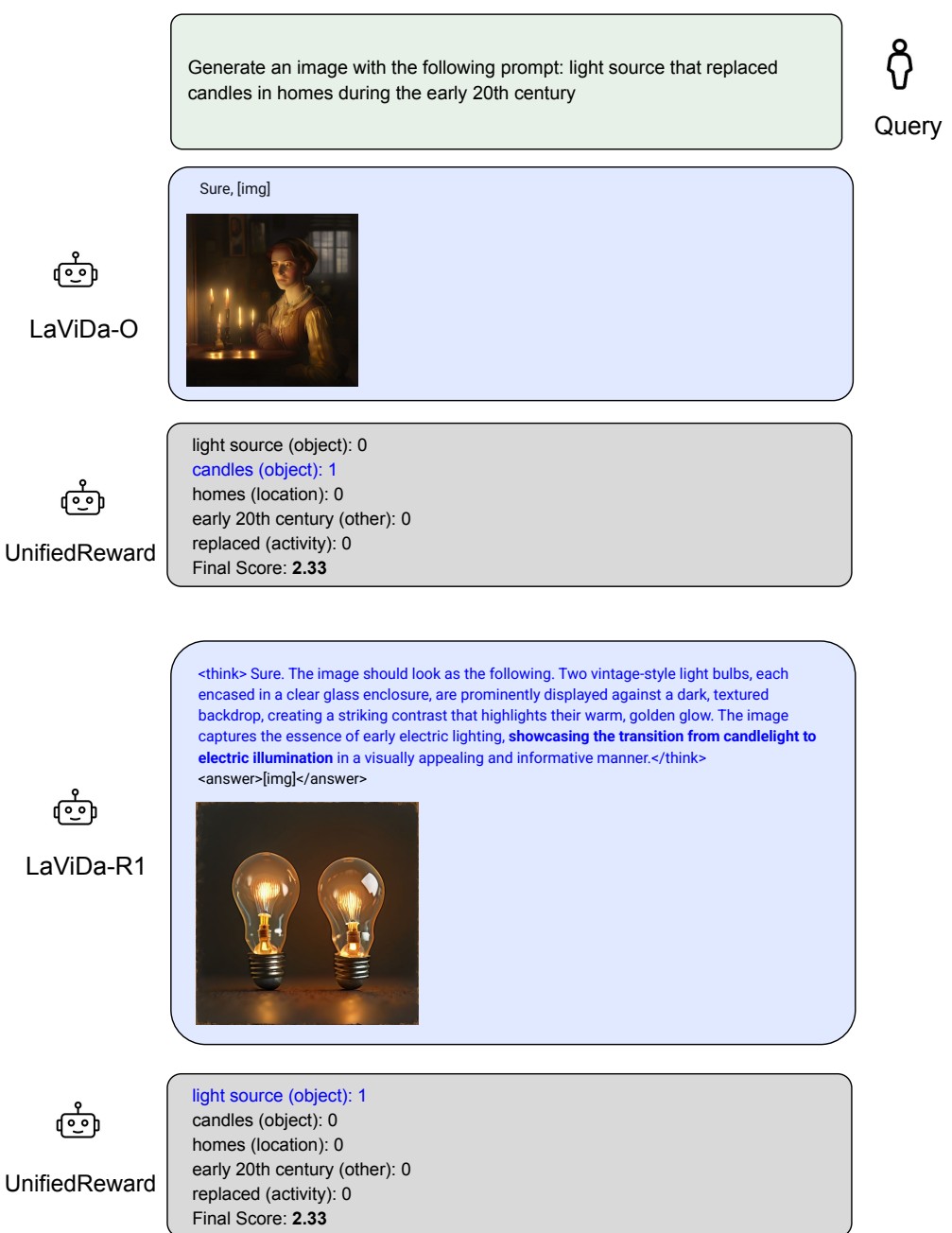

Figure 11. **Qualitative Reuslts on Text-to-Image Generation.** While LaViDa-R1 demonstrated some zero-shot reasoning capabilities on text-to-image tasks, we find that existing VLM-based reward models fail to properly provide a reward signal for reasoning-based tasks.

