# OpenReview forum: "Lavida-R1: Advancing Reasoning for Unified Multimodal Diffusion Language Models"
_ICML.cc/2026/Conference — ICML 2026 regular_

### Official Review · Reviewer_GDFM · 2026-03-10

**Soundness:** 2
**Presentation:** 3
**Significance:** 3
**Originality:** 3
**Overall Recommendation:** 3
**Confidence:** 2

**Summary:**

This paper introduces LaViDa-R1, a unified post-training framework designed to enhance the reasoning capabilities of multimodal diffusion language models. The authors reveal that while dLLMs excel in parallel generation, they struggle with complex logical tasks and suffer from training instability during reinforcement learning due to intractable likelihood estimation. To address this, LaViDa-R1 employs a Unified Post-training objective that replaces traditional KL-divergence with SFT regularization, preventing model collapse while encouraging broader exploration. Additionally, the authors propose Guided Rollout strategies—Answer-Forcing and Tree Search—to generate high-quality training signals for reasoning-intensive tasks. Experimental results across diverse benchmarks demonstrate that LaViDa-R1 significantly outperforms existing dLLMs and competitive autoregressive models in both multimodal understanding and generative reasoning.

**Compliance With Llm Reviewing Policy:**

Affirmed.

**Final Justification:**

My concerns are partially addressed and I'm actually not very familiar with this field.

**Key Questions For Authors:**

- Necessarily of Tree Search and Self-Distillation. From Table 5 and Table 7, I found the improvements are quite limited on ImgEdit. From my understanding, the Tree Search introduce a lot of additional cost, which raise the concerns that whether these designs are necessary.
- Limited evaluation. The authors claim a general-purpose reasoning dLLM. However, the evaluation is limited to
five benchmarks, which reduce the reliability of the such a strong claim.
- Question about the motivation for the Unified Post-training. The authors mentions that the KL divergence in RL for AR models is not suitable for dLLMs. However, there several RL algorithm that do not use KL divergence, such as DAPO. I think the Clip-Higher in DAPO serves  as a strong baseline for the Unified Post-training.

**Limitations:**

The authors adequately discuss the limitations.

**Strengths And Weaknesses:**

- This paper explores an important research problem of diffusion MLLM.
- This paper is well-written and easy to follow.
- The experiments demonstrate the effectiveness of proposed methods.

---

> ### Author Rebuttal · Authors · 2026-03-31
>
> **Q1 Necessity of Tree Search and Self-Distillation**
>
> We address this concern from two perspectives: performance gain and compute cost.
>
>
> **Performance Gain:** It is important to contextualize the gains in terms of the scales of benchmark scores. First we note that the base model LaViDa-O achieves a score of 3.71 without thinking and 3.80 (+0.09) with thinking. LaVida-R1 further boosts this number to 3.90 (+0.19), this means the “effectiveness of thinking” is more improved by 110% (+0.09 &rarr; +0.19). Among these additional +0.10 gains, Table 5 shows that 50% of them come from the proposed tree search algorithm (3.85&rarr;3.90), and that such gain cannot be achieved by naively scaling compute. Hence,  tree search algorithm plays a significant role in the overall improvement
>
> We also want to note that many other works demonstrated similar scales of post training gains on ImgEdit, including UniWorld-V2 [1] (+0.13), ReasonEdit [2] (+0.12), FIRM-Qwen-Edit [3] (+0.07).
>
> **Compute Overhead.**  We provide additional analysis on the computational overhead. Given a fixed sequence length, inference FLOPs is proportional to the product of batch size and number of function evaluations (NFE). We report this number as normalized FLOPs. For example, generating one image with 64 steps or two images with 32 steps will both have normalized FLOPs of 64. We report Table 5 results with additional FLOPs information. We also included a new setup (Exp #4) with group size 8x2 and tree search step [0,8]. Comparing Exp 1 and 4, Exp 2 and 5, Exp 3 and 6, we find that the proposed tree search algorithm always reduces the compute cost compared with i.i.d sampling to obtain the same total number of samples, this is because in tree search, we can skip early diffusion steps.
>
> | Exp Id | Tree Search Step | Group Size | ImgEdit | FLOPs (Normalized) |
> | --- | --- | --- | --- | --- |
> | 1 | N/A | 16 | 3.85 | 1024 |
> | 2 | N/A | 32 | 3.84 | 2048 |
> | 3 | N/A | 64 | 3.84 | 4096 |
> | 4 | [0,8] | 8x2 | 3.89 | 960 |
> | 5 | [0,8] | 16x2 | 3.90 | 1920 |
> | 6 | [0,8,16,32] | 16x4 | 3.87 | 3200 |
>
>
> **Q2: Results only reported on 5 benchmarks.**
>
> We appreciate the concern about evaluation breadth. To clarify, our evaluation spans 9 distinct benchmarks. Table 1 already included 7 benchmarks for understanding tasks. Table 2 included performance on ImgEdit and Table 3 included object grounding tasks. It is important to note that Figure 1 only visualize the performance of 5 representative benchmarks out of these 9 for better visual clarity.
>
> As requested by a different reviewer DHkj, we also incorporated 3 more benchmarks MMMU, MME and MMBench in the rebuttal. These results are included below
>
> |  | Capabilities | MMMU | MMB | MME |
> | --- | --- | --- | --- | --- |
> | LaViDa-O | Und & Gen | 45.1  | 76.4 | 488 |
> | Lavida-R1 | Und & Gen | 47.0(+1.9) | 79.2 (+2.8) | 501 (+13) |
>
> **Q3: Comparison with DAPO.**
>
> We note that the clip-higher design is not applicable to LaViDa-R1 because our training is fully on-policy. This means that DAPO,GRPO, and other variants which differ on the clipping policy are equivalent in our setup because $\mu=1$ and $\pi_{old}$ is just $\pi_{\theta}$. Specifically for DAPO,  clipping in their Eq 9 will not be triggered, and DAPO and GRPO are equivalent.
>
> In general, methods like DAPO mostly focus on finding a better clipping strategy to address the distribution gap of $\pi_{old}$ and $\pi_{\theta}$, which can be eliminated with a fully on-policy training where  $\pi_{old}=\pi_{\theta}$ and clipping is never triggered.
>
>  [1] Li, Z., Liu, Z., Zhang, Q., Lin, B., Wu, F., Yuan, S., Yan, Z., Ye, Y., Yu, W., Niu, Y. and Wang, S., 2025. Uniworld-v2: Reinforce image editing with diffusion negative-aware finetuning and mllm implicit feedback. arXiv preprint arXiv:2510.16888.
>
> [2] Yin, F., Liu, S., Han, Y., Wang, Z., Xing, P., Wang, R., Cheng, W., Wang, Y., Li, A., Yin, Z. and Chen, P., 2025. ReasonEdit: Towards Reasoning-Enhanced Image Editing Models. arXiv preprint arXiv:2511.22625.
>
> [3] Zhao, X., Zhang, P., Lin, J., Liang, T., Duan, Y., Ding, S., Tian, C., Zang, Y., Yan, J. and Yang, X., 2026. Trust Your Critic: Robust Reward Modeling and Reinforcement Learning for Faithful Image Editing and Generation. arXiv preprint arXiv:2603.12247.

---

> > ### Author Rebuttal · Reviewer_GDFM · 2026-04-01
> >
> > Thanks the authors for the reply. My concerns are partially resolved.

---

### Official Review · Reviewer_DHkj · 2026-03-12

**Soundness:** 3
**Presentation:** 3
**Significance:** 3
**Originality:** 3
**Overall Recommendation:** 5
**Confidence:** 4

**Summary:**

The paper proposes LaViDa-R1, a multimodal general-purpose reasoning dLLM. LaviDa-R1 unifies diverse multimodal understanding and generation task with a novel post-training framework which integrates SFT and multi-task RL. During the post-training, the authors employ several techniques such as answer-forcing, tree search and complimentary likelihood estimation to have even better results.The result shows strong performance on reasoning, grounding and image editing.

**Compliance With Llm Reviewing Policy:**

Affirmed.

**Final Justification:**

The rebuttal has addressed all my concerns and I am willing to increase the score to 5.

**Key Questions For Authors:**

Please see weaknesses above. I believe the paper in its current form already presents promising ideas+results.

**Limitations:**

Please see weaknesses above.

**Strengths And Weaknesses:**

### Strengths
1. The paper is well-written and easy to follow with clear figures that illustrate the proposed techniques quite well.
2. The unified policy gradient formulation (GRPO, SFT, best-of-n distillation) that expands on different tasks (visual math reasoning, image editing and grounding) is elegant.
2. The unified model achieves competitive performance on a variety of task like math reasoning, grounding and image editing.

### Weakness
1. For the VQA comparison, more comparison should be included for better comparison, some VQA benchmarks like MME and MMMU from the [MMaDA](https://openreview.net/pdf?id=wczmXLuLGd) paper. Additionally,  [LLaDA-V](https://arxiv.org/pdf/2505.16933) reports some of the same benchmarks used in this paper but are not listed in the evaluation tables.
2. The paper mentions EditScore evaluation being a major bottleneck for post training. Especially for table 5, group size = 16 with no tree search (3.85) and [0,8]  (3.90), the difference seems relatively small. It would be helpful to include qualitative comparisons of generated outputs and report the corresponding training FLOPs or compute cost. Providing this information would clarify the trade-off between performance improvement and computational expense and would be useful for future work studying post-training efficiency.

---

> ### Author Rebuttal · Authors · 2026-03-30
>
> **W1: VQA Benchmarks**
>
> As requested, we report results on more benchmarks like MME and MMMU with additional baselines. We will include these results in the final version.
>
> |  | Capabilities | MMMU | MMB | MME |
> | --- | --- | --- | --- | --- |
> | LLaDa-V | Und Only | 48.6 | 82.9 | 491 |
> | MMaDa | Und & Gen | 30.2 | 68.5 | 242 |
> | LaViDa-O | Und & Gen | 45.1  | 76.4 | 488 |
> | Lavida-R1 | Und & Gen | 47.0(+1.9) | 79.2 (+2.8) | 501 (+13) |
>
>
> **W2: Performance Gains are limited on ImgEdit.**
>
> We address this concern from two perspectives: performance gain and compute cost.
>
>
> **Performance Gain:** It is important to contextualize the gains in terms of the scales of benchmark scores. First we note that the base model LaViDa-O achieves a score of 3.71 without thinking and 3.80 (+0.09) with thinking. LaVida-R1 further boosts this number to 3.90 (+0.19), this means the “effectiveness of thinning” is more improved by 110%  (+0.09 &rarr; +0.19). Among these additional +0.10 gains, Table 5 shows that 50% of them come from the proposed tree search algorithm (3.85&rarr;3.90), and that such gain cannot be achieved by naively scaling compute. Hence, the tree search algorithm plays a significant role in the overall improvement
>
> We also want to note that many other works demonstrated similar scales of post training gains on ImgEdit, including UniWorld-V2 [1] (+0.13), ReasonEdit [2] (+0.12), FIRM-Qwen-Edit [3] (+0.07).
>
> **Compute Overhead.**  We provide additional analysis on the computational overhead. Given a fixed sequence length, inference FLOPs is proportional to the product of batch size and number of function evaluations (NFE). We report this number as normalized FLOPs. For example, generating one image with 64 steps or two images with 32 steps will both have normalized FLOPs of 64. We report Table 5 results with additional FLOPs information. We also included a new setup (Exp #4) with group size 8x2 and tree search step [0,8]. Comparing Exp 1 and 4, Exp 2 and 5, Exp 3 and 6, we find that the proposed tree search algorithm always reduces the compute cost compared with i.i.d sampling to obtain the same total number of samples, this is because in tree search, we can skip early diffusion steps.
>
> | Exp Id | Tree Search Step | Group Size | ImgEdit | FLOPs (Normalized) |
> | --- | --- | --- | --- | --- |
> | 1 | N/A | 16 | 3.85 | 1024 |
> | 2 | N/A | 32 | 3.84 | 2048 |
> | 3 | N/A | 64 | 3.84 | 4096 |
> | 4 | [0,8] | 8x2 | 3.89 | 960 |
> | 5 | [0,8] | 16x2 | 3.90 | 1920 |
> | 6 | [0,8,16,32] | 16x4 | 3.87 | 3200 |
>
> Further, we would like to clarify that the Editscore overhead is mostly on the memory side as we need extra servers to host the reward model. However, these servers have very low average GPU utilization at 3%, and most of the compute time is used to generate samples for RL training. We do not need to provision more reward servers when changing the group size or adding the tree search, as they are under-utilized.
>
>
> [1] Li, Z., Liu, Z., Zhang, Q., Lin, B., Wu, F., Yuan, S., Yan, Z., Ye, Y., Yu, W., Niu, Y. and Wang, S., 2025. Uniworld-v2: Reinforce image editing with diffusion negative-aware finetuning and mllm implicit feedback. arXiv preprint arXiv:2510.16888.
>
> [2] Yin, F., Liu, S., Han, Y., Wang, Z., Xing, P., Wang, R., Cheng, W., Wang, Y., Li, A., Yin, Z. and Chen, P., 2025. ReasonEdit: Towards Reasoning-Enhanced Image Editing Models. arXiv preprint arXiv:2511.22625.
>
> [3] Zhao, X., Zhang, P., Lin, J., Liang, T., Duan, Y., Ding, S., Tian, C., Zang, Y., Yan, J. and Yang, X., 2026. Trust Your Critic: Robust Reward Modeling and Reinforcement Learning for Faithful Image Editing and Generation. arXiv preprint arXiv:2603.12247.

---

> > ### Author Rebuttal · Reviewer_DHkj · 2026-03-31
> >
> > Both of my concerns have been addressed and I am willing to improve the score to accept

---

### Official Review · Reviewer_ivpx · 2026-03-13

**Soundness:** 4
**Presentation:** 3
**Significance:** 3
**Originality:** 2
**Overall Recommendation:** 4
**Confidence:** 4

**Summary:**

The paper introduces LaViDa-R1, a general-purpose, multimodal reasoning diffusion language model (dLLM). Moving away from the paradigm of task-specific reinforcement learning, the authors propose a unified post-training framework tailored for dLLMs that seamlessly integrates supervised fine-tuning (SFT), multi-task reinforcement learning (RL), and self-distillation objectives. To enhance both scalability and effectiveness, the framework incorporates several training techniques, including answer-forcing, tree search, and complementary likelihood estimation. Extensive experiments demonstrate that LaViDa-R1 achieves strong, benchmark-leading performance across a wide array of multimodal tasks, such as visual math reasoning, reason-intensive grounding, and image editing.

**Compliance With Llm Reviewing Policy:**

Affirmed.

**Final Justification:**

The authors have largely addressed my concerns, and I will maintain my positive score.

**Key Questions For Authors:**

Please see weaknesses above.

**Limitations:**

yes

**Strengths And Weaknesses:**

### Strengths
1. Effectiveness: Introduces a highly effective unified post-training framework for multimodal dLLMs.
2. Technical Execution: Successfully integrates SFT, RL, and self-distillation, notably addressing and preventing the training collapse often faced by standard GRPO.
3. Empirical Results: Achieves benchmark-leading performance across a wide range of complex multimodal tasks.

### Weaknesses
1. Limited Novelty: The framework reads more like an engineering combination of prevalent post-training approaches rather than a fundamentally novel methodology.
2. Incremental Sub-components: Techniques highlighted as novel, such as "answer-forcing" and "tree search," are standard ideas heavily utilized in prior NLP literature.
3. Inaccurate Terminology: The use of "tree search" is misleading, as no actual tree structure is built or traversed. A term like "stratified Monte-Carlo sampling" would be mathematically more accurate.

---

> ### Author Rebuttal · Authors · 2026-03-30
>
> **W1 Limited Novelty**
>
> We thank the reviewer for the comment. We agree that the individual ideas such as RL with GRPO are not entirely new for LLM/VLMs. However, we want to emphasize that applying these ideas to multimodal dLLMs in multitask setting introduces challenges that do not exist in the language-only or AR settings, and our specific design choices are driven by these challenges.
>
>  For example, Figure 5 shows that training can diverge even with KL regularizer, highlighting the need for our proposed mixture of SFT and RL loss. We observed that this challenge uniquely occurs for the multi-modal setting with a mixture of image and text tokens, since image tokens have higher uncertainty (measured by cross entropy loss) than text tokens, which causes instability in the KL estimators. We also note that unified multi-task RL for dLLMs is an under-explored area, and many existing works only focus on single-task optimization in the language domain. Designing a unified post-training framework that works for a wide range of multi-modal tasks and reward signals is not trivial.
>
> **W2 Incremental Sub-components**
>
> We note that answer-forcing and tree search are novel in the sense that they are most natural in the context of dLLMs with any order generation capability and parallel diffusion generation process. To the best of our knowledge, many of these concepts are only recently discussed in concurrent works.
>
> For example, answer-forcing is not possible for an AR model which has a causal attention mask and a strictly left-to-right generation order. We cannot fix the answer at the end to generate reasoning in the middle. Similar concepts in language domain were only recently explored in a concurrent work [1] (Jan 26) for AR LLMs. However, it focuses on a different domain (continual learning) and has to design a convoluted system prompt to ask the AR LLM to generate the reasoning trace conditioned on a given final answer. To the best of our knowledge, no prior work applied similar ideas to the sampling process of RL training.
>
> We respectfully ask the reviewer to point to some specific literature. We are happy to include them as additional discussions to better highlight the novelty of LaViDa-R1.
>
> **W3 Terminology**
>
> We thank the reviewers for the suggestion and will revise the term “tree search” as suggested in the final version to better reflect the method. For consistency with the paper and readability for AC and other reviewers, we apologize for keeping it for the moment in the above response and response to other reviewers.
>
> [1] Shenfeld, Idan, et al. "Self-Distillation Enables Continual Learning." arXiv preprint arXiv:2601.19897 (2026).

---

> > ### Author Rebuttal · Reviewer_ivpx · 2026-04-03
> >
> > Thank you for the rebuttal. The authors have largely addressed my concerns, and I will maintain my positive score.

---

### Official Review · Reviewer_2XpP · 2026-03-13

**Soundness:** 3
**Presentation:** 2
**Significance:** 3
**Originality:** 2
**Overall Recommendation:** 4
**Confidence:** 3

**Summary:**

LaViDa-R1 proposes a training recipe for improving reasoning in unified multimodal diffusion LLMs. The core idea is pretty clean: they unify SFT, online GRPO-style RL, and best-of-N self-distillation into a single weighted policy-gradient framework. They add two guided rollout mechanisms (answer forcing and tree search) plus a complementary-masking likelihood estimator for stable gradients. Results across visual reasoning, text-only reasoning, image editing, and grounding show consistent gains over their LaViDa-O base, with especially strong improvements on grounding and text-only reasoning.

**Compliance With Llm Reviewing Policy:**

Affirmed.

**Key Questions For Authors:**

1. How is the restart timestep t_s selected (heuristics, schedules, adaptive criteria)? Do you observe reduced diversity when always branching from the top sample?
2. Can you add controlled comparisons against d1/UniGRPO and report results for other methods like VLM-R1, SATORI, or OneThinker on overlapping benchmarks? This would help situate your gains better.

**Limitations:**

yes

**Strengths And Weaknesses:**

**Strengths**
1. The unified post-training formulation is elegant—combining SFT, on-policy RL (without KL), and best-of-N self-distillation through the same policy-gradient interface with per-sample weights is practical and clean.
2. Answer-forcing is clever. It leverages the bidirectional/inpainting property of dLLMs to synthesize high-quality reasoning traces conditioned on ground-truth answers when exploration fails. This is a nice use of the dLLM generative process that you can't do with standard autoregressive models.
3. The evaluation is broad, covering both understanding and generation tasks (including reason-intensive grounding and image editing). This shows the approach transfers across modalities and reward types, which is reassuring.
4. This demonstrates that dLLMs can be improved for multimodal reasoning with an integrated, stable post-training pipeline—a useful complement to existing AR-focused R1-style methods.

**Weaknesses**
1. The comparisons are limited. There aren't enough head-to-head comparisons with closely related "R1-style" multimodal RL approaches and unified RL methods (e.g., UniGRPO/d1 on the same base, VLM-R1, OneThinker) on overlapping benchmarks. This makes it hard to contextualize the absolute gains.
2. The tree search ablation shows small, sometimes negligible gains. More analysis would help—like diversity metrics, depth/branch-factor trade-offs—to clarify when it actually matters.
3. Several critical training details (task mixture ratios, reward scaling per task, group sizes per task) are deferred to the appendix; the main text would benefit from a concise summary for reproducibility.

---

> ### Author Rebuttal · Authors · 2026-03-30
>
> We thank the reviewer for the overall positive comments.
>
>
> **W1: Limited Comparison.**
>
> We note that UniGRPO/d1 are not directly comparable since they focused on task-specific finetuning as opposed to unified multitask learning. However, we did provide ca omparison for the UniGRPO/d1-style loss using the same base and multitask dataset in Table 6. As noted in Sec 5 line 413-424, these experiments are equivalent to the UniGRPO loss with MC=1, MC=2 and d1 loss.  We include a copy of Table 6 with this information as below
>
> | #MC | Masking | LISA-Grounding | ImgEdit | Equiv. To |
> | --- | --- | --- | --- | --- |
> | 1 | i.i.d | 61.9 | 3.82 | UniGRPO (MC=1) |
> | 1 | Full | 59.2 | 3.77 | d1 |
> | 2 | i.i.d | 62.1 | 3.86 | UniGRPO (MC=2) |
> | 2 | Compl. | 65.0 | 3.88 | Ours |
>
> This equivalence is also explained in Appendix Figure 6.
>
> For VLM-R1, it focuses on object grounding tasks and is already included in Table 3 for comparison. For other works like OneThinker, they use stronger AR base models (Qwen-VL Series) and are not directly comparable. The current version of Table 1 focuses on dLLMs. We will include a new section in Table 1 dedicated to AR models and their RL finetunes in the final version for completeness.
>
> **W2 Small gains for TreeSearch (Table 5)**
>
> It is important to contextualize the gains in terms of the scales of benchmark scores. First, we note that the base model LaViDa-O achieves a score of 3.71 on ImgEdit without thinking and 3.80 (+0.09) with thinking. LaVida-R1 further boosts this number to 3.90 (+0.19), this means the “effectiveness of thinking” is improved by 110% (+0.09 &rarr; +0.19). Among these additional +0.10 gains, Table 5 shows that 50% of them come from the proposed tree search algorithm (3.85&rarr;3.90), and that such gain cannot be achieved by naively scaling compute. Hence, tree search algorithm plays a significant role that is not “negligible”.
>
> We also would like to note that many other works demonstrated similar scales of post-training gains on ImgEdit, including UniWorld-V2 [1] (+0.13), ReasonEdit [2] (+0.12), FIRM-Qwen-Edit [3] (+0.07).
>
> [1] Li, Z., Liu, Z., Zhang, Q., Lin, B., Wu, F., Yuan, S., Yan, Z., Ye, Y., Yu, W., Niu, Y. and Wang, S., 2025. Uniworld-v2: Reinforce image editing with diffusion negative-aware finetuning and mllm implicit feedback. arXiv preprint arXiv:2510.16888.
>
> [2] Yin, F., Liu, S., Han, Y., Wang, Z., Xing, P., Wang, R., Cheng, W., Wang, Y., Li, A., Yin, Z. and Chen, P., 2025. ReasonEdit: Towards Reasoning-Enhanced Image Editing Models. arXiv preprint arXiv:2511.22625.
>
> [3] Zhao, X., Zhang, P., Lin, J., Liang, T., Duan, Y., Ding, S., Tian, C., Zang, Y., Yan, J. and Yang, X., 2026. Trust Your Critic: Robust Reward Modeling and Reinforcement Learning for Faithful Image Editing and Generation. arXiv preprint arXiv:2603.12247.
>
> **W3 The main text would benefit from a concise summary for reproducibility.**
>
> We appreciate the suggestion and will include a summary of training details from the appendix into the main text in the final version.
>
> **Q.1a How is t_s selected.**
>
> We select t_s based on two steps. First, we narrow down possible candidates to {8,16,32,48} based on the heuristic that small changes in t_s will probably not lead to large changes in the results (e.g. 8 vs 9), this avoids the need to test out all 64 timesteps. We then conduct empirical experiments and visual validations to test these choices. We include the results below
>
>
> |  | Group Size | ImgEdit |
> | --- | --- | --- |
> | [0,8] | 16x2 | 3.90 |
> | [0,16] | 16x2 | 3.88 |
> | [0,32] | 16x2 | 3.85 |
> | [0,48] | 16x2 | 3.85 |
>
> Visually, we find that branching from later steps leads to less diversity because later steps have fewer mask tokens and the partially generated image is “more complete”. In particular, branching from 48-th diffusion step leads to samples that are almost identical visually, and have small differences in reward scores.
>
> **Q1.b Do you observe reduced diversity when always branching from the top sample.**
>
> As discussed above, since we branch from early diffusion steps with sufficient mask tokens left to be generated, the generated samples are diverse and have meaningful differences in reward scores, which is necessary to make the tree search work. We will include visual examples in the final version. Per ICML policy, we cannot include new images at this stage since there is no rebuttal PDF.
>
> **Q2 Add controlled comparisons for d1/UniGRPO**
>
> (See W1)

---

> > ### Author Rebuttal · Reviewer_2XpP · 2026-04-05
> >
> > The authors' rebuttal has addressed most of my concerns, I will keep my score.

---

### Decision · Program_Chairs · 2026-04-30

**Decision:**

Accept (regular)

**Comment:**

This paper introduces Lavida-R1, a unified post-training framework for multimodal diffusion language models. It seamlessly integrates supervised fine-tuning, multi-task RL, and self-distillation, while introducing novel guided rollout techniques like answer-forcing and tree search to improve reasoning capabilities across visual math, grounding, and image editing tasks.

Initial concerns primarily centered around limited baselines, the computational overhead/necessity of the tree search component, and some terminology choices. Following the rebuttal, three reviewers are in favor of acceptance. The remaining negative reviewer failed to engage in follow-up discussions.